# RANKFLOW: PROPERTY-AWARE TRANSPORT FOR PROTEIN OPTIMIZATION

**Lu Yu**[1,2]**, Wei Xiang**[1,2,†]**, Kang Han**[1]**, Gaowen Liu**[3]**, and Ramana Rao Kompella**[3]

[1] School of Computing, Engineering and Mathematical Sciences, La Trobe University, Melbourne, Australia
[2] Australian Centre for AI and Medical Innovation, La Trobe University, Melbourne, Australia
[3] Cisco Research, San Jose, CA
{l.yu,w.xiang, k.han}@latrobe.edu.au,{gaoliu, rkompell}@cisco.com

## ABSTRACT

A key step in protein optimization is modeling the fitness landscape, which maps proteins to functional assay readouts. Existing methods typically either use property-agnostic likelihoods/embeddings from pretrained protein language models (PLMs) for fitness prediction, or assume independent mutational effects, limiting their ability to capture higher-order interactions. In this work, we introduce RankFlow, a conditional flow framework that refines PLM representations to be a property-aligned distribution via a tailored energy function and captures multi-mutation interactions through learnable embeddings. To align optimization with evaluation protocols, we propose the Rank-Consistent Conditional Flow Loss ($RC^2$), a differentiable ranking objective that enforces the correct order of mutants rather than absolute values, which improves out-of-distribution generalization. Finally, we introduce a Property-guided Steering Gate (PSG) that concentrates learning on positions carrying signals for the target property while suppressing unrelated evolutionary biases. Across the ProteinGym, PEER, and FLIP benchmarks, RankFlow obtains state-of-the-art ranking accuracy and superior generalization performance.

## 1 INTRODUCTION

Proteins enable catalysis, structural support, molecular transport, and cellular signaling. The ability to design protein sequences and structures with tailored properties has a direct impact on sustainable manufacturing and therapeutics. In practice, optimization targets a series of properties that can include stability, binding affinity, and enzymatic activity, which are typically measured by high-throughput assays (Biswas et al., 2021). A central objective is to learn property landscapes that map variants to these assay readouts and that explain how single and combined mutations modulate them (Notin et al., 2024). The more accurately we model and navigate such landscapes, the more reliably we can propose edits that deliver targeted behavior in realistic experimental settings.

Protein optimization typically begins by modeling the mapping from sequence and structure to functional readouts, often called the *fitness landscape* (Romero & Arnold, 2009). A significant obstacle in modeling the fitness landscape is the scarcity of experimentally measured labels. This has motivated self-supervised representation learning for mutation effect prediction (Hopf et al., 2017; Meier et al., 2021). Early approaches, including Riesselman et al. (2018); Frazer et al. (2021); Laine et al. (2019) have modeled family-specific distributions from multiple sequence alignments (MSAs) and used those priors to score variants. More recently, hybrid designs have combined family-specific and family-agnostic information for fitness prediction (Alley et al., 2019; Rives et al., 2021). Those PLMs have shown that their predicted likelihoods can infer evolutionary trajectories and predict zero-shot mutation effects (Rao et al., 2021; Notin et al., 2022). Subsequently, with the growth of high-throughput assays and broader annotated datasets, many studies have moved to supervised training (Gelman et al., 2021; Heinzinger et al., 2019). They fine-tune PLMs on experimental readouts and minimize differences between predicted and measured property values.

---

†Corresponding Author

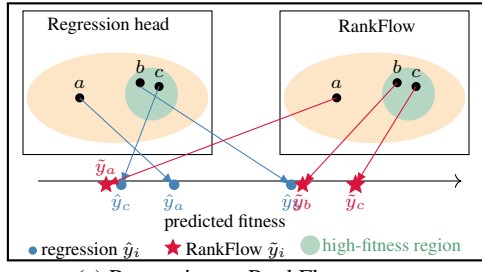 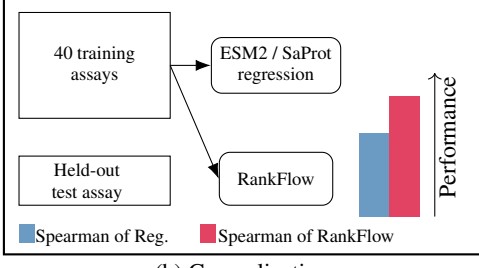

(a) Regression vs RankFlow        (b) Generalization

Figure 1: Illustration of RankFlow. (a) In a local region of representation space, a regression head on PLM embeddings can overfit when fine-tuned on a single assay, mapping some truly high-fitness mutants to low scores and vice versa. RankFlow instead reshapes wild-type-conditioned mutant representations into a fitness-aligned distribution, enforcing a property-aware landscape. (b) In a cross-assay generalization experiment, models are trained on 40 Deep Mutational Scanning (DMS) assays from the same category and evaluated on a held-out assay; RankFlow achieves higher Spearman correlation than fine-tuned ESM2/SaProt with regression heads, indicating stronger generalization.

However, two critical aspects remain underaddressed. First, while PLMs capture rich evolutionary regularities, PLM representations are property-agnostic (Loell & Nanda, 2018): they encode multiple, often competing evolutionary constraints (e.g., foldability, stability, expression), which can dilute or even oppose the signal of the property of interest. Directly using such embeddings can bias predictions toward wild-type-like preferences rather than property-aligned improvements. Second, some fitness prediction methods overlook multi-mutation interactions (epistasis) and assume additive effects of individual mutations (Notin et al., 2023a; Wang et al., 2024). This leads to wrong predictions, especially for higher-order mutants where non-additive interactions dominate and significantly influence the target property (Dauparas et al., 2022; Chen et al., 2024).

In this work, we present RankFlow, a property-aware conditional flow framework designed to address both limitations while preserving the generalization ability of pretrained PLMs. Standard approaches attach a regression head to PLM embeddings and fine-tune on a single DMS assay, which can yield strong in-assay performance but tends to introduce dataset-specific bias and inferior cross-protein generalization. Rather than performing point-to-point regression, RankFlow keeps the PLM embeddings as a base sequence/structure prior and learns an energy-guided conditional flow that reshapes wild-type-conditioned mutant representations into a fitness-aligned distribution. This refinement captures multi-mutation interactions through learnable embeddings defined on mutation sets, going beyond additive assumptions. The energy function is constructed from observed fitness scores and steers the learned dynamics toward landscapes where high-fitness mutants consistently outrank low-fitness ones. Figure 1 illustrates how RankFlow differs from regression methods. To align training with the evaluation protocols of protein engineering, we propose the Rank-Consistent Conditional Flow Loss ($RC^2$), a differentiable soft-ranking objective that enforces the correct order of mutants rather than absolute values, improving robustness to noise and generalization to unseen assays. Finally, we introduce a Property-guided Steering Gate (PSG), which focuses learning on positions carrying signals for the target property and suppresses directions reflecting unrelated evolutionary biases, reducing wild-type preference and sharpening the edit signal. We evaluate RankFlow on ProteinGym (Notin et al., 2023a), $\beta$-Lactamase and Fluorescence from PEER (Xu et al., 2022), and GB1 from FLIP (Dallago et al., 2021), where it consistently outperforms state-of-the-art methods in ranking accuracy and generalization. Our work makes the following contributions:

- We propose RankFlow, a novel conditional flow framework that refines PLM-derived mutant representations into property-aligned embeddings, capturing non-additive interactions across mutation sets.

- We introduce the Rank-Consistent Conditional Flow Loss ($RC^2$ Loss), a rank-consistent flow objective that aligns training with rank-based evaluation metrics and improves out-of-distribution generalization.

- We develop the Property-guided Steering Gate (PSG) mechanism, which focuses learning on property-relevant positions and reduces wild-type bias in PLM representations.

## 2 RELATED WORK

**Protein Representation Learning.** Protein representation learning has seen significant advancements with the introduction of protein language models (PLMs) that leverage large-scale protein sequence data (Rives et al., 2021). Early works focused on family-specific models trained on MSAs (Riesselman et al., 2018; Hopf et al., 2017; Laine et al., 2019), which capture evolutionary constraints within protein families. More recent approaches have developed family-agnostic PLMs (Rao et al., 2021; Lin et al., 2023; Yang et al., 2023) that learn from vast and diverse protein sequences, enabling zero-shot prediction of mutational effects and improved generalization across different proteins. Hybrid models that combine family-specific and family-agnostic information have also been proposed (Alley et al., 2019; Notin et al., 2022; Hsu et al., 2022; Su et al., 2024), further enhancing the predictive capabilities of PLMs. These models typically use self-supervised objectives, such as masked language modeling or autoregressive prediction, to learn rich representations that can be fine-tuned for specific downstream tasks.

**Protein Fitness Prediction.** Protein fitness prediction aims to model the relationship between proteins and their functional properties, such as stability and activity (Dallago et al., 2021; Gelman et al., 2021). Traditional approaches directly map sequences to measured properties via regression or classification (Gelman et al., 2021; Heinzinger et al., 2019), while alignment-based models focus on specific protein families and use evolutionary information from MSAs to infer mutational effects (Riesselman et al., 2018; Laine et al., 2019; Shin et al., 2021). Unsupervised methods predict mutation effects by evaluating changes in likelihood scores (Riesselman et al., 2018; Frazer et al., 2021; Notin et al., 2022; Nijkamp et al., 2023; Truong Jr & Bepler, 2023), and more recent work leverages PLM representations within supervised frameworks for improved fitness prediction (Wang et al., 2024; Notin et al., 2023b; Tan et al., 2025). Despite these advances, existing methods primarily optimize for absolute fitness regression, which can limit robustness when extrapolating to unseen mutation combinations. Notin et al. (2023b) introduces a non-parametric Transformer that jointly embeds protein sequences and labels and excels in label-scarce settings. Beck et al. (2025) propose a meta-learning approach that trains an in-context regressor over more than 100 fitness tasks, but it assumes access to many related tasks and excludes proteins longer than 750 amino acids due to memory limits. Both methods are also computationally and memory-intensive during training and inference. Kermut (Groth et al., 2024) combines sequence and structure information in a Gaussian process with composite kernels and achieves state-of-the-art performance; however, it inherits the cubic scaling of exact GPs and thus becomes costly on large variant libraries, and requires truncation or subsampling for long or dense assays (Groth et al., 2024). Ronen et al. (2025) further shows that Kermut and related predictors can be sensitive to the choice of representation and use prediction-based screening plus representation ensembling to stabilize performance and improve out-of-distribution generalization and uncertainty quantification. In contrast, our method learns a lightweight, property-aware flow on top of PLM representations, and explicitly models multi-mutation interactions to enable strong generalization even on large and diverse variant libraries.

## 3 METHOD

This section describes the proposed RankFlow framework. Section 3.1 presents the problem setup and model overview. Section 3.2 introduces the Property-aware Conditional Flow, which constructs a new property-guided flow to generate the target distribution. In Section 3.3, we describe the Steering Gate. Section 3.4 details the RankFlow architecture and training and inference procedures.

### 3.1 PROBLEM SETUP AND MODEL OVERVIEW

Protein optimization seeks to quantify how sequence edits change a property of interest. Let a wild type protein be $x^{\text{wt}} = [x_1^{\text{wt}}, \ldots, x_N^{\text{wt}}]$ with $N$ amino acids (AAs). A mutation is a set of substitutions $\mu = \{\mu_m : x_m^{\text{wt}} \rightarrow x_m^{\text{mt}} \mid m \in \{1, \ldots, M\}\}$, which may act on multiple positions at once. Applying $\mu$ yields a mutant sequence $x^{\text{mt}} = [x_1^{\text{mt}}, \ldots, x_N^{\text{mt}}]$. For a wild type protein and a given assay, a typical learning objective is to fit a function $\mathcal{F}_{\theta}(x^{\text{wt}}, \mu) = y$, where $y$ is the measured property value of the mutation $\mu$ on $x^{\text{wt}}$.

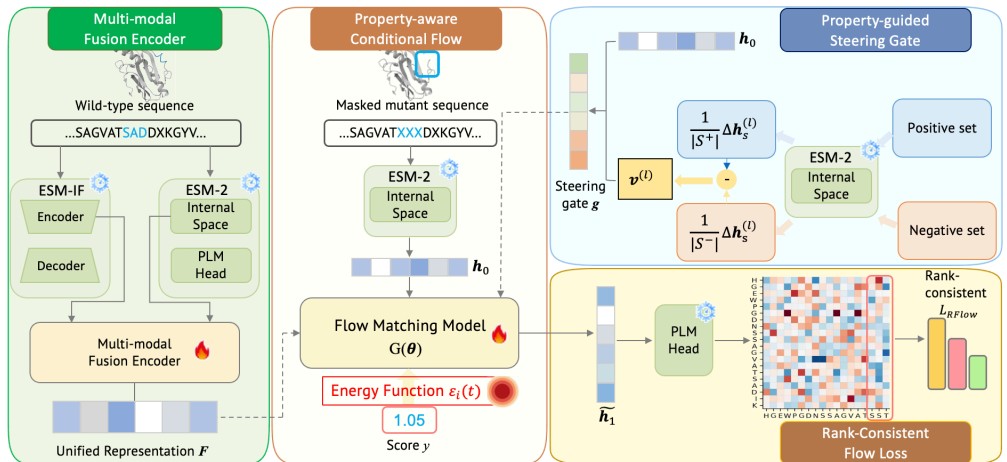

Figure 2: Overview of RankFlow for protein fitness prediction. RankFlow takes wild-type-conditioned PLM representations as the source distribution $p_0$ and learns a property-aware conditional flow (Section 3.2) that transports $p_0$ to a property-aligned target $q$. The flow is conditioned on the wild type context (via the Multi-modal Fusion Encoder), the mutation set, and a Property-guided Steering Gate (Section 3.3) that emphasizes positions relevant to the target property. Training uses a dual-objective loss that combines property-aware flow matching with the rank-consistent $RC^2$ loss.

In RankFlow, rather than fitting a deterministic predictor $\mathcal{F}_\theta$ that is prone to overfitting in the low-data regime, we train a conditional flow parameterized by $\theta$ that models a distribution over protein representations whose density is tilted toward the target property via our energy function. This flow takes PLM representations as the source distribution $p_0$ and transports it to a property-aware distribution $q$ conditioned on $\mu$ and $y$. Specifically, for each mutant $x_i^{\mathrm{mt}}$ in the dataset, we mask the mutated AAs with the PLM's mask token, feed the sequence into the PLM, and extract the final hidden representation $h_0 \in \mathbb{R}^{N \times d}$ prior to the output head. Masking removes self-information about the wild type residues, yielding a mutation-aware yet wild-type-specific state $h_0 \sim p_0$, which serves as the input distribution for RankFlow. The overview of RankFlow is depicted in Figure 2.

## 3.2 PROPERTY-AWARE CONDITIONAL FLOW

**Continuous Property-aware Flow.** Inspired by Zhang et al. (2025), we learn a conditional flow that generates samples not directly from the base distribution $p_0$, but from an *energy-guided* distribution $q(h) \propto p_0(h) \exp\{-\mathcal{E}(h)\}$, where $\mathcal{E}(h)$ is an energy function that encodes the target property. Following guided flows (Zheng et al., 2023), we model a conditional distribution $p_t(h \,|\, h_0)$ at time $t$ via a Gaussian conditional path:

$$p_t(h \,|\, h_0) \;=\; \mathcal{N}(\mu_t\, h_0,\; \sigma_t^2 I), \tag{1}$$

where $\mu_t, \sigma_t$ are differentiable scheduler functions. The property-aware distribution at time $t$ is:

$$q_t(h) \;\propto\; p_t(h)\, \exp\{-\mathcal{E}_t(h)\}, \tag{2}$$

with $\mathcal{E}_t$ denotes the energy at time $t$. To realize this distribution, we construct a flow via a time-dependent velocity field $v_t(\theta)$ describing the velocity of the particle at position $h$ that matches the conditional vector field $u_t(h \,|\, h_0)$. For the Gaussian path $p_t(h \,|\, h_0)$, this field admits the closed form with score function $\log p_t(h \,|\, h_0)$:

$$u_t(h \,|\, h_0) = \dot{\mu}_t\, \mu_t^{-1}\, h \;+\; (\dot{\mu}_t\, \sigma_t - \mu_t\, \dot{\sigma}_t)\, \sigma_t\, \mu_t^{-1}\, \nabla_h \log p_t(h \,|\, h_0), \tag{3}$$

where $\dot{\mu}_t$ and $\dot{\sigma}_t$ are both the derivative of $\mu_t$ and $\sigma_t$ with respect to time t.

**Energy function.** We construct an assay-invariant energy function that emphasizes mutants with strong fitness scores while remaining robust to differences across assays. Our design combines two principles: (*i*) mutants with large fitness values should be favored, and (*ii*) mutants whose fitness

departs from local substitution patterns should be highlighted (Wylie & Shakhnovich, 2011). Let $\mathcal{K}$ be the training set ($K = |\mathcal{K}|$). Mutant $\boldsymbol{x}_i^{\mathrm{mt}}$ has measured label $y_i$ and a mutation set $\boldsymbol{\mu}_i = \{(k, a_{i,k})\}$ where $k$ denotes a mutated site and $a_{i,k}$ the amino acid at that site. We measure similarity between two mutants $\boldsymbol{x}_i^{\mathrm{mt}}$ and $\boldsymbol{x}_j^{\mathrm{mt}}$ by a substitution-aware edit distance:

$$d_{\mathrm{sub}}(i,j) = \left| S_i \triangle S_j \right| + \left| \{\, k \in S_i \cap S_j : a_{i,k} \neq a_{j,k} \,\} \right|, \quad S_i = \{\, k : (k, a_{i,k}) \in \boldsymbol{\mu}_i \,\}, \quad (4)$$

so that $d_{\mathrm{sub}}(i,j) = 1$ means $\boldsymbol{x}_i^{\mathrm{mt}}$ and $\boldsymbol{x}_j^{\mathrm{mt}}$ differ by a single mutational move. For each mutant $\boldsymbol{x}_i^{\mathrm{mt}}$, we define its neighborhood $\mathcal{N}(i) = \{\, j \in \mathcal{K} \mid j \neq i,\ d_{\mathrm{sub}}(i,j) \leq r \,\}$, where $r$ is a mutational radius. Within this neighborhood, we compute kernel weights $K_{ij}$ and normalize them:

$$K_{ij} = \exp\big(-d_{\mathrm{sub}}(i,j)\big), \quad \widehat{K}_{ij} = \frac{K_{ij}}{\sum_{j \in \mathcal{N}(i)} K_{ij}}. \quad (5)$$

Using the neighborhood weights, we then compute:

$$\bar{\tilde{y}}_i = \sum_{j \in \mathcal{N}(i)} \widehat{K}_{ij}\,\tilde{y}_j, \qquad s_i = \sum_{j \in \mathcal{N}(i)} \widehat{K}_{ij}\,(\tilde{y}_j - \bar{\tilde{y}}_i)^2, \quad (6)$$

where $\tilde{y}_i$ is the normalized fitness score with $\tilde{y}_i = (y_i - \mu_{\mathrm{tr}})/(\sigma_{\mathrm{tr}})$, where $\mu_{\mathrm{tr}}$ and $\sigma_{\mathrm{tr}}$ are the mean and deviation. $\bar{\tilde{y}}_i$ is the local baseline implied by nearby mutants, and $s_i$ is the corresponding variance. Finally, with flow time $t \in [0, 1]$, we form the energy function $\mathcal{E}_i(\boldsymbol{h})$ and weight $w_i(t)$ by combining a global magnitude term and a local deviation term:

$$\mathcal{E}_i(\boldsymbol{h}) = -\big(\lambda\,\tilde{y}_i + (1-\lambda)\,\frac{\tilde{y}_i - \bar{\tilde{y}}_i}{\sqrt{s_i}}\big), \qquad w_i(t) \propto \exp\{-\beta \mathcal{E}_i(\boldsymbol{h})\}, \quad (7)$$

where $\tilde{y}_i$ denotes the standardized global fitness of variant $\boldsymbol{x}_i^{\mathrm{mt}}$, encouraging the flow to shift embeddings toward globally high-fitness regions, $(\tilde{y}_i - \bar{\tilde{y}}_i)/\sqrt{s_i}$ quantifies how much mutant $i$ deviates from local substitution patterns. The mixing coefficient $\lambda$ balances these two components, and the final weight $w_i(t)$ emphasizes mutants that are simultaneously high in standardized fitness and anomalous relative to local substitution trends. $\beta > 0$ controls the sharpness of the weighting.

**Rank-Consistent Conditional Flow Loss.** With energy weights $w_i(t)$, we learn the flow by matching the vector field $\boldsymbol{v}_t(\boldsymbol{\theta})$ to the target velocity $\boldsymbol{u}_t(\boldsymbol{h} \mid \boldsymbol{h}_0)$:

$$\mathcal{L}_{\mathrm{PFM}}(\boldsymbol{\theta}) = \mathbb{E}_{t,\boldsymbol{h},\boldsymbol{h}_0}\left[\tilde{w}_i(t)\,\|\boldsymbol{v}_t(\boldsymbol{h};\boldsymbol{\theta}) - \boldsymbol{u}_t(\boldsymbol{h} \mid \boldsymbol{h}_0)\|_2^2\right], \quad (8)$$

where $\boldsymbol{v}_t$ is the vector field. $\tilde{w}_i(t)$ is obtained by applying a softmax to $w_i(t)$. Given infinite data, $\mathcal{L}_{\mathrm{PFM}}(\boldsymbol{\theta})$ is able to achieve the global minimum. However, in our task, we observe a main limitation with this objective: some assays contain only a few hundred labeled variants, which is insufficient to learn a complex transport map that accurately matches property values.

Therefore, we propose a Rank-Consistent Conditional Flow Loss (RC$^2$) that complements the $\mathcal{L}_{\mathrm{PFM}}(\boldsymbol{\theta})$. Specifically, for each sample $\boldsymbol{x}_i^{\mathrm{mt}}$, the representation $\boldsymbol{h}_0$ from the PLM's internal space is fed into the flow model $G(\boldsymbol{\theta})$ to produce a predicted target representation $\tilde{\boldsymbol{h}}_1$. We compute target logits with the PLM head, $\tilde{Q}^{\mathrm{tgt}} = \mathrm{PLMHead}(\tilde{\boldsymbol{h}}_1)$, and form the predicted score $\tilde{y}$ by summing logit differences over the mutated sites $\boldsymbol{\mu}_i$:

$$\tilde{y}_i \simeq \sum_{m \in \boldsymbol{\mu}_i} \left(\log \tilde{Q}^{\mathrm{tgt}}_{m = \boldsymbol{x}_m^{\mathrm{mt}}} - \log \tilde{Q}^{\mathrm{tgt}}_{m = \boldsymbol{x}_m^{\mathrm{wt}}}\right). \quad (9)$$

$\tilde{y}_i$ can be regarded as the predicted property value for the mutant $\boldsymbol{x}_i^{\mathrm{mt}}$. This readout is invariant to the scale of logits and focuses on the mutations that drive the property change (Gordon et al., 2025). For a batch of $B$ mutants, let $\tilde{\boldsymbol{y}} = \{\tilde{y}_i\}$ denote the model predictions and $\boldsymbol{y} = \{y_i\}$ be the experimental measured labels. Our proposed RC$^2$ loss $\mathcal{L}_{\mathrm{RFlow}}(\boldsymbol{\theta})$ minimizes a differentiable surrogate of the Spearman rank correlation between $\tilde{\boldsymbol{y}}$ and $\boldsymbol{y}$:

$$\mathcal{L}_{\mathrm{RFlow}}(\boldsymbol{\theta}) = \lambda_{\mathrm{rank}}\big(1 - \rho_{\mathrm{soft}}\big(\mathrm{R}_\tau(\tilde{\boldsymbol{y}}), \mathrm{R}(\boldsymbol{y})\big)\big), \quad (10)$$

where $\lambda_{\mathrm{rank}}$ is a hyper-parameter, $\mathrm{R}_\tau(\cdot)$ is a standard differentiable ranking operator used in prior work (Cuturi et al., 2019) on differentiable sorting and ranking with temperature $\tau > 0$, the hard rank operator $R(\cdot)$ is used for ground-truth labels, and $\rho_{\mathrm{soft}}$ computes a differentiable correlation between (soft) predicted ranks and the hard ranks of target $\boldsymbol{y}$. Now the total loss is defined as:

$$\mathcal{L}(\boldsymbol{\theta}) = \mathcal{L}_{\mathrm{PFM}}(\boldsymbol{\theta}) + \mathcal{L}_{\mathrm{RFlow}}(\boldsymbol{\theta}). \quad (11)$$

This total loss combines the property-aware flow matching and the rank-wise flow objectives, balancing the need to align with the target property while preserving the correct ranking of mutants.

### 3.3 REPRESENTATION OPTIMIZATION VIA A PROPERTY-GUIDED STEERING GATE

While the training objective Eq. 11 directly optimizes representations for a target property, the evolutionary information from PLMs is property-agnostic. They encode many objectives at once and can steer updates toward directions that are neutral or even adverse for a given assay (Notin et al., 2022). To address this issue, we propose a Property-guided Steering Gate (PSG) that concentrates learning on positions carrying signals for the target property and suppresses directions driven by unrelated evolutionary biases.

Let $\mathcal{S}^+$ and $\mathcal{S}^-$ be the top and bottom $\xi$-quantiles of the training set by the measured property (we use $\xi = 0.3$ unless stated). For mutant $\boldsymbol{x}_i^{\mathrm{mt}}$, we define the wild-type-conditioned token delta by:

$$\Delta \boldsymbol{h}_i^{(\ell)} \;=\; \boldsymbol{h}^{(\ell)}(\boldsymbol{x}_i^{\mathrm{mt}}) \;-\; \boldsymbol{h}^{(\ell)}(\boldsymbol{x}^{\mathrm{wt}}), \tag{12}$$

where $\boldsymbol{h}^{(\ell)} \in \mathbb{R}^{N \times d}$ denotes the token representation from $\ell$-layer of PLM. Empirically, we use the final PLM layer ($\ell = L$) from the PLM's internal space before its prediction head. This delta ties the signal at each position to the effect of mutating that site and reduces wild-type-specific offsets. We then compute average per-position token representations across sequences in each set and construct a steering matrix $\boldsymbol{V}_l$ that points along the mean difference between high- and low-property samples:

$$\boldsymbol{\mu}_+^{(\ell)} = \frac{1}{|\mathcal{S}^+|} \sum_{s \in \mathcal{S}^+} \Delta \boldsymbol{h}_s^{(\ell)}, \qquad \boldsymbol{\mu}_-^{(\ell)} = \frac{1}{|\mathcal{S}^-|} \sum_{s \in \mathcal{S}^-} \Delta \boldsymbol{h}_s^{(\ell)}, \tag{13}$$

$$\boldsymbol{V}^{(\ell)} \;=\; \boldsymbol{\mu}_+^{(\ell)} - \boldsymbol{\mu}_-^{(\ell)}. \tag{14}$$

$\boldsymbol{V}^{(\ell)}$ captures the direction that separates high- and low-property mutants in the PLM representation space. Prior to training, we compute $\boldsymbol{V}^{(\ell)}$ once and cache it. For each mutant $\boldsymbol{x}_i^{\mathrm{mt}}$ during training, we score position $n$ by a cosine similarity to $\boldsymbol{V}^{(\ell)}$:

$$w_{i,n} = \frac{\langle \Delta \boldsymbol{h}_{i,n}^{(\ell)}, \boldsymbol{V}_n^{(\ell)} \rangle}{\|\Delta \boldsymbol{h}_{i,n}^{(\ell)}\| \, \|\boldsymbol{V}_n^{(\ell)}\| + \varepsilon}, \tag{15}$$

where $\Delta \boldsymbol{h}_{i,n}^{(\ell)}$ is the $n$-th row of $\Delta \boldsymbol{h}_i^{(\ell)}$, $\varepsilon > 0$ is a small constant for numerical stability. Large positive $w_{i,n}$ indicates the token's representation aligns with the high-property direction, while negative $w_{i,n}$ indicates opposition. We convert these scores to a gate vector:

$$\boldsymbol{g}_i \;=\; \gamma \, \sigma(\boldsymbol{w}_i), \tag{16}$$

with scale $\gamma > 0$ and sigmoid $\sigma(\cdot)$. This gate serves as a condition to our flow model to concentrate learning on positions that carry signal for the target property and reduces the influence of positions that reflect unrelated evolutionary signals.

### 3.4 RANKFLOW ARCHITECTURE

The implementation of RankFlow comprises two core components: (1) Multi-modal fusion encoder, which combines structure and sequence representations for a protein sequence; and (2) Conditional flow head: which predicts the velocity field at each time step.

**Multi-modal Fusion Encoder.** We leverage existing state-of-the-art architectures to encode the structure and sequence modalities separately. For structure encoding, we rely on the pretrained structure encoder ESM-IF (Hsu et al., 2022), which can effectively capture the geometric context of the wild-type protein. For sequence encoding, we utilize pretrained PLM ESM-2 (Lin et al., 2023) to extract evolutionary information from wild-type sequences. The outputs from both encoders are projected using two Multi-Layer Perceptrons (MLPs) and then fused via a self-attention block to create a unified representation $\boldsymbol{F} \in \mathbb{R}^{N \times d}$ that captures both sequence and structural features.

**Conditional Flow Head.** The conditional flow head is designed to predict the velocity field $\boldsymbol{v}_t(\cdot)$ at each time step $t$. To refine PLM representations, we add learnable embeddings to the flow head, applied only at mutated sites with one embedding per position. For a mutation set $\boldsymbol{\mu} = \{\mu_n\}_{m=1}^M$, we construct $\boldsymbol{c}(\boldsymbol{\mu}) \in \mathbb{R}^{L \times d_c}$ with entries $\boldsymbol{c}_m(\boldsymbol{\mu}) = [\phi_{\mathrm{pos}}(m) + \phi_{\mathrm{aa}}(\mu_m)]$, where $\phi_{\mathrm{pos}} \in \mathbb{R}^{L \times d_c}$

and $\phi_{\mathrm{aa}} \in \mathbb{R}^{20 \times d_c}$ are trainable modules. $\boldsymbol{c}_m(\boldsymbol{\mu})$ will be concatenated to $\boldsymbol{h}_0$ so that the flow can learn mutation-specific adjustments. The flow head takes as input the current state $\boldsymbol{h}_t$, the mutation set $\boldsymbol{\mu}$, and the condition $\boldsymbol{C}$, which includes the fused representation $\boldsymbol{F}$ and the steering gate $\boldsymbol{g}$. The conditional flow head is now parameterized as $\boldsymbol{v}_t(\boldsymbol{h}|\boldsymbol{C};\boldsymbol{\theta})$, where $\boldsymbol{\theta}$ is a light stack of U-Net blocks with time embedding and layer normalization to effectively model the dynamics of the flow.

Putting all the components together, RankFlow is trained as a conditional flow-matching model. For each mutant $x_i^{\mathrm{mt}}$, we first obtain its representation $\boldsymbol{h}_i^0$ from the frozen PLM, together with the fused sequence-structure representation $\boldsymbol{F}$ and steering gate $\boldsymbol{g}_i$ obtained from Eq. 16; these are collected into the condition $\boldsymbol{C}_i$. We then sample a time $t \sim \mathcal{U}(0,1)$, construct a noisy state $\boldsymbol{h}_t = \mu_t \boldsymbol{h}^0 + \sigma_t \varepsilon$ with fixed scheduler $(\mu_t, \sigma_t)$ and $\varepsilon \sim \mathcal{N}(0, I)$, and compute the target velocity $\boldsymbol{u}_t(\boldsymbol{h}_t \,|\, \boldsymbol{h}^0)$ in closed form according to Eq. 3, where we omit the index $i$ for brevity. The flow head takes $(\boldsymbol{h}_t, \boldsymbol{C})$ as input and predicts a velocity $\boldsymbol{v}_t(\boldsymbol{h}_t; \theta)$. The parameters $\theta$ are optimized by minimizing the energy-weighted flow-matching loss $L_{\mathrm{PFM}}$ in Eq. 8 together with the rank-consistent loss in Eq. 10. Algorithm 1 in Appendix A.1 summarizes the overall training procedure. At inference time, we fix $\boldsymbol{C}$ for a given assay and mutant, integrate the learned vector field from $t = 1$ to $t = 0$ using a fixed-step Heun solver with $N$ steps (Appendix A.2), and map the final representation to a scalar fitness score.

## 4 EXPERIMENT

### 4.1 IMPLEMENTATION DETAILS

We first fix the architecture and optimiser, then select method-specific hyperparameters on a few representative assays. For the loss composition, we start from $L_{\mathrm{PFM}}$ and add $\mathcal{L}_{\mathrm{RFlow}}$ only if it improves validation Spearman (which it consistently does), so we recommend using the combined loss. For the energy design of $\lambda$, we perform a coarse sweep 0,0.25,0.5,1 and choose the value near the validation optimum (typically $\lambda = 0.5$). For scheduling, we compare linear vs. cosine time schedules and pick the better one (cosine in our case). Once chosen on this validation subset, we reuse the same configuration for all ProteinGym assays without per-assay tuning; full training and evaluation configurations are given in Appendix B.6. We use Spearman's rank correlation coefficient between predicted and experimentally measured fitness values as the evaluation metric.

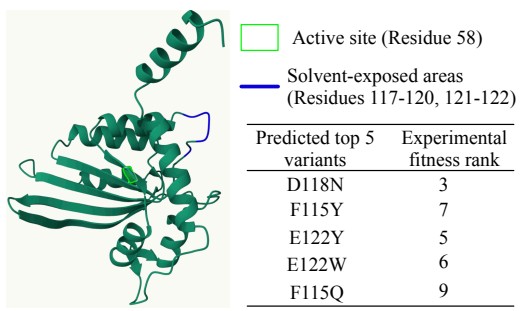

| Predicted top 5 variants | Experimental fitness rank |
|---|---|
| D118N | 3 |
| F115Y | 7 |
| E122Y | 5 |
| E122W | 6 |
| F115Q | 9 |

Active site (Residue 58)

Solvent-exposed areas (Residues 117-120, 121-122)

Figure 3: Structural visualization of AICDA_HUMAN using Mol* Viewer (Sehnal et al., 2021). The top three predicted mutants by RankFlow are at solvent-exposed positions.

### 4.2 PERFORMANCE COMPARISON

**Benchmarks.** ProteinGym (Notin et al., 2023a) serves as a benchmark for mutation effect prediction. It contains 217 substitution DMS assays and over 2.4 million mutated sequences spanning more than 200 protein families. Additional details are provided in Appendix B.1. $\beta$-lact. and Fluo. are widely used benchmarks for protein fitness prediction. Both comprise deep mutational scanning measurements covering thousands of single- and multi-mutants. $\beta$-lact. includes 4,158 training, 520 validation, and 520 test samples; Fluo. contains 21,446 training, 5,362 validation, and 27,217 test samples. GB1 from FLIP (Dallago et al., 2021) is derived from deep mutational scanning of the IgG-binding protein G domain B1, which assayed nearly all single to quadruple mutants. In the 2-vs-rest split, double mutants are used for training, while single, triple, and quadruple mutants are held out for validation and testing.

Table 1 reports the performance of RankFlow and other baselines on four benchmarks. For the ProteinGym benchmark, we follow DePLM (Wang et al., 2024) in excluding datasets whose wild-type proteins exceed 1,024 residues, yielding 201 DMS datasets. Among these methods,

Table 1: Spearman performance of supervised methods on $\beta$-lact., GB1, Fluo., and ProteinGym under the Random scheme. Results of CNN, ResNet, LSTM, and Transformer are from Wang et al. (2024); OHE, ESM-MSA, Tranception, and ProteinNPT are from Notin et al. (2023b). ▨: best, ▨: second best.

| Model | ProteinGym | | | | | $\beta$-lact. | GB1 | Fluo. |
|---|---|---|---|---|---|---|---|---|
| | Stability | Fitness | Expression | Binding | Activity | | | |
| CNN | 0.788 | 0.588 | 0.627 | 0.599 | 0.573 | 0.781 | 0.502 | 0.682 |
| ResNet | 0.734 | 0.489 | 0.521 | 0.525 | 0.481 | 0.152 | 0.133 | 0.636 |
| LSTM | 0.745 | 0.413 | 0.477 | 0.496 | 0.408 | 0.139 | -0.002 | 0.494 |
| Transformer | 0.560 | 0.149 | 0.156 | 0.172 | 0.155 | 0.261 | 0.271 | 0.643 |
| OHE | 0.718 | 0.545 | 0.573 | 0.562 | 0.555 | 0.823 | 0.533 | 0.657 |
| ESM-1v | 0.880 | 0.566 | 0.642 | 0.596 | 0.572 | 0.536 | 0.394 | 0.438 |
| ESM-2 | 0.882 | 0.573 | 0.645 | 0.587 | 0.576 | – | – | – |
| ESM-MSA | 0.885 | 0.568 | 0.632 | 0.565 | 0.600 | – | – | – |
| ProtSSN | 0.877 | 0.692 | 0.718 | 0.757 | 0.678 | – | – | – |
| SaProt | 0.882 | 0.686 | 0.716 | 0.749 | 0.677 | – | – | – |
| Tranception | 0.871 | 0.632 | 0.704 | 0.671 | 0.623 | – | – | – |
| ProteinNPT | 0.904 | 0.668 | 0.736 | 0.706 | 0.680 | – | – | – |
| DePLM (ESM1v) | 0.887 | 0.704 | 0.738 | 0.773 | 0.688 | 0.900 | 0.676 | 0.662 |
| DePLM (ESM2) | 0.897 | 0.707 | 0.742 | 0.764 | 0.693 | 0.904 | 0.665 | 0.662 |
| RankFlow (Ours) | 0.911 | 0.742 | 0.765 | 0.781 | 0.722 | 0.912 | 0.689 | 0.687 |

ProtSSN (Tan et al., 2025), SaProt Su et al. (2024), DePLM (Wang et al., 2024), and our model are sequence+structure approaches that use both amino acid sequences and 3D structures, whereas all other methods operate on sequence information only. All models are trained either from scratch or via task-specific fine-tuning. As shown in Table 1, our RankFlow achieves the state-of-the-art performance on all benchmarks consistently. Both ESM-MSA (Rao et al., 2021) and Tranception (Notin et al., 2022) outperform ESM-1v (Meier et al., 2021) and ESM-2 (Lin et al., 2023) by leveraging evolutionary information from MSAs. However, RankFlow still outperforms them significantly, demonstrating that reorganizing information in pretrained PLMs can surpass even MSA-based approaches. Compared to DePLM, which also uses a generative model, RankFlow directly learns a conditional flow that transports property-agnostic PLM embeddings to a property-aligned distribution. By modeling mutation sets rather than independent sites, it better captures higher-order epistatic interactions that DePLM can only approximate, and consistently surpasses DePLM across benchmarks. We provide a per-assay comparison of Spearman scores in Fig. 8. We further analyzed which variants are prioritized by our method and found that, as shown in Fig. 3, high-fitness predictions are enriched at solvent-exposed positions and away from the active site, which is consistent with established biological observations (Notin et al., 2022).

We further evaluate RankFlow on ProteinGym under the Contiguous and Modulo schemes, with results summarized in Table 2. Kermut (Groth et al., 2024) achieves the best performance on the Contiguous split, where a single long region of positions is entirely withheld during training. In contrast, RankFlow achieves the highest Spearman scores on both the Random and Modulo splits. The Modulo split suppresses periodic subsets of positions while still preserving distributed positional coverage; RankFlow's conditional-flow formulation appears to leverage this dispersed contextual information more effectively than kernel-based regression. Aggregated over all three evaluation modes, RankFlow yields the highest average performance. This suggests that RankFlow exhibits more stable performance across different forms of distribution shift.

## 4.3 UNCERTAINTY ESTIMATION

We adopt the hybrid uncertainty estimation used in ProteinNPT, combining MC-Dropout and batch-resampling to obtain per-mutation epistemic uncertainty scores. This approach is model-agnostic and does not require any changes to our model architecture. As reported in Table 3, the Spearman correlation between our uncertainty scores and the absolute prediction errors ($\rho_{\text{uncertainty}}$) is comparable to or in some schemes slightly higher than the values reported for Stable CNN, Stable Bayesian Ridge, and Stable Kermut (Ronen et al., 2025). These results indicate that our approach not only achieves strong predictive accuracy but also delivers robust uncertainty quantification on par with stabilized baselines.

Table 2: Spearman performance on the ProteinGym benchmark under Random, Modulo, and Contiguous evaluation schemes. Results for Kermut are taken from the original paper (Groth et al., 2024), while all other baseline results are sourced from the ProteinGym website.

| Model | Random | Modulo | Contiguous | Avg. | Std. err. |
|---|---|---|---|---|---|
| OHE | 0.582 | 0.022 | 0.059 | 0.221 | 0.014 |
| ESM-1v + OHE | 0.565 | 0.394 | 0.396 | 0.452 | 0.015 |
| DeepSequence + OHE | 0.523 | 0.393 | 0.395 | 0.437 | 0.017 |
| MSAT + OHE | 0.540 | 0.409 | 0.407 | 0.452 | 0.014 |
| Tranception + OHE | 0.550 | 0.413 | 0.415 | 0.459 | 0.012 |
| TranceptEVE + OHE | 0.552 | 0.435 | 0.435 | 0.474 | 0.012 |
| ESM-1v Emb. | 0.614 | 0.514 | 0.479 | 0.535 | 0.014 |
| MSAT Emb. | 0.670 | 0.562 | 0.513 | 0.581 | 0.013 |
| Tranception Emb. | 0.681 | 0.525 | 0.439 | 0.548 | 0.008 |
| ProteinNPT | 0.741 | 0.588 | 0.529 | 0.619 | 0.009 |
| Kermut | 0.744 | 0.631 | 0.591 | 0.655 | 0.000 |
| RankFlow (Ours) | 0.786 | 0.635 | 0.589 | 0.669 | 0.018 |

Table 3: Benchmark results on ProteinGym under the Contiguous and Modulo evaluation schemes. $\rho_{uncertainty}$ is the Spearman correlation between the uncertainty scores and the absolute errors of the predictions. All baseline model values are taken from Ronen et al. (2025).

| Model | Contiguous | | Modulo | | Random | |
|---|---|---|---|---|---|---|
| | Spearman | $\rho_{uncertainty}$ | Spearman | $\rho_{uncertainty}$ | Spearman | $\rho_{uncertainty}$ |
| CNN | 0.344±0.013 | 0.019±0.006 | 0.344±0.013 | 0.019±0.006 | 0.365±0.012 | 0.013±0.006 |
| Stable CNN | 0.492±0.010 | 0.129±0.008 | 0.492±0.010 | 0.129±0.008 | 0.509±0.010 | 0.133±0.008 |
| Bayesian Ridge | 0.422±0.014 | 0.008±0.004 | 0.422±0.014 | 0.008±0.004 | 0.693±0.013 | 0.005±0.004 |
| Stable Bayesian Ridge | 0.597±0.011 | 0.153±0.008 | 0.597±0.011 | 0.153±0.008 | 0.755±0.011 | 0.139±0.007 |
| Kermut | 0.606 ± 0.012 | 0.110±0.008 | 0.606±0.012 | 0.110±0.008 | 0.758±0.012 | 0.106±0.008 |
| Stable Kermut | 0.667 ± 0.010 | 0.164±0.010 | 0.667 ± 0.010 | 0.164 ± 0.010 | 0.785 ± 0.010 | 0.184 ± 0.008 |
| RankFlow (Ours) | 0.589±0.009 | 0.135 ± 0.009 | 0.635 ± 0.010 | 0.132 ± 0.008 | 0.786 ± 0.012 | 0.165 ± 0.006 |

## 4.4 GENERALIZATION

To demonstrate the generalization of RankFlow, we further test it on the ProteinGym benchmark (Notin et al., 2023a). Following the experimental settings in DePLM (Wang et al., 2024), for each test dataset, we randomly select 40 additional datasets from the same category for training, while ensuring that sequence similarity between training and test datasets remains below 50% to avoid data leakage. Table 4 summarizes the performance of RankFlow and the baselines on ProteinGym. RankFlow consistently outperforms all methods across categories, indicating strong generalization. Despite using PLMs, it has far fewer trainable parameters than SaProt (37.1M vs. 650M), enabling training within about 1 hour on a single A100 GPU, whereas some large-scale baselines require days. This efficiency and data efficiency make RankFlow well suited for protein engineering applications.

## 4.5 ABLATION STUDY

We conduct an ablation study to analyze the contributions of RankFlow's components, comparing the full model with four variants: (1) energy-guided conditional flow only (RankFlow ($\mathcal{L}_{PFM}$ only)); (2) RC$^2$ loss only (RankFlow ($\mathcal{L}_{RFlow}$ only)); (3) RankFlow w/o structure info; and (4) RankFlow w/o steering gate $g_i$. We evaluate all variants on ProteinGym, and report the results by (a) function type and (b) mutation depth. As shown in Fig. 4, our energy-guided objective $\mathcal{L}$PFM yields the largest gains across assays and mutation depths, underscoring the importance of property-aware flow matching for learning an accurate fitness landscape. The RC$^2$ loss is especially helpful at higher mutation depths, where the combinatorial explosion of sequences makes reliable supervision scarce. Removing the property-guided steering gate (RankFlow w/o steering gate $g_i$) improves over the RC$^2$-only variant, highlighting the value of focusing learning on positions most relevant to the target property. We also observe a moderate performance drop when structure information and the multi-modal fusion encoder are removed (RankFlow w/o Structure info), particularly on label-sparse

Table 4: Generalization ability evaluation on ProteinGym. Results for zero-shot and fine-tuned (FT) baselines are taken from Wang et al. (2024).

| Model | Input Modalities | | Trainable | Spearman | | | | |
|---|---|---|---|---|---|---|---|---|
| | Seq. | Struct. | parameters | Stability | Fitness | Expression | Binding | Activity |
| *Zero-shot methods* | | | | | | | | |
| ESM1v | ✓ | | - | 0.437 | 0.395 | 0.427 | 0.287 | 0.415 |
| ESM2 | ✓ | | - | 0.523 | 0.396 | 0.439 | 0.356 | 0.433 |
| ProtSSN | ✓ | ✓ | - | 0.560 | 0.408 | 0.435 | 0.362 | 0.458 |
| TranceptEVE L | ✓ | | - | 0.500 | 0.477 | 0.457 | 0.360 | 0.487 |
| ESM-IF | | ✓ | - | 0.624 | 0.346 | 0.436 | 0.380 | 0.412 |
| ProteinMPNN | | ✓ | - | 0.564 | 0.166 | 0.209 | 0.159 | 0.203 |
| *Supervised methods* | | | | | | | | |
| CNN | | ✓ | 2.7M | 0.141 | 0.053 | 0.043 | 0.056 | 0.095 |
| ESM1v (FT) | ✓ | ✓ | 650M | 0.497 | 0.318 | 0.301 | 0.216 | 0.385 |
| ESM2 (FT) | ✓ | ✓ | 650M | 0.454 | 0.359 | 0.338 | 0.276 | 0.391 |
| ProtSSN (FT) | ✓ | ✓ | 148M | 0.689 | 0.448 | 0.478 | 0.421 | 0.488 |
| SaProt (FT) | ✓ | ✓ | 650M | 0.703 | 0.442 | 0.496 | 0.391 | 0.495 |
| DePLM (ESM1v) | ✓ | ✓ | 42.2M | 0.763 | 0.467 | 0.506 | 0.409 | 0.499 |
| DePLM (ESM2) | ✓ | ✓ | 42.2M | 0.773 | 0.480 | 0.510 | 0.441 | 0.518 |
| **RankFlow (Ours)** | ✓ | ✓ | 37.1M | **0.797** | **0.515** | **0.534** | **0.457** | **0.554** |

ProteinGym assays (notably in the Fitness and Activity categories), yet the sequence-only variant still surpasses strong sequence-only baselines such as ESM2(FT) and SaProt(FT), indicating that RankFlow remains effective without structures. More details are in Appendix B.6.

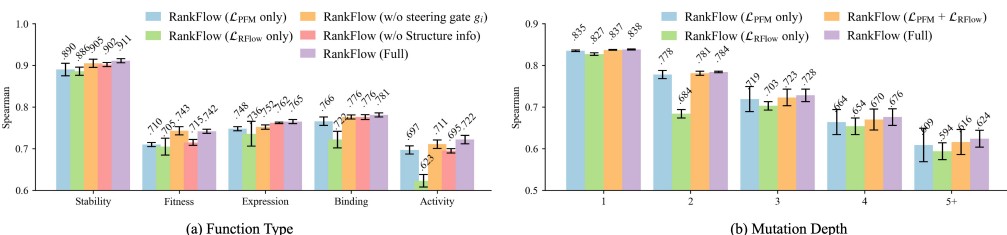

Figure 4: Results of ablation study for analyzing contributions of different components. Breakdown performance on assays is grouped by (a) function type and (b) mutation depth.

## 5 CONCLUSION

We present RankFlow, a conditional flow-matching framework for protein fitness prediction that learns a property-aware landscape on top of pretrained PLMs. By introducing an energy function tied to assay-specific fitness and a rank-consistent objective, it shapes the flow so that mutants are ordered coherently by their functional properties. In addition, the property-guided steering gate focuses learning on relevant positions, improving performance on diverse protein engineering tasks. Across ProteinGym and additional protein engineering benchmarks, RankFlow consistently matches or surpasses state-of-the-art supervised methods under the same training protocols, using far fewer trainable parameters than full PLM fine-tuning and offering a robust, transferable approach to protein fitness prediction.

**Reproducibility Statement.** All the datasets used in this work are publicly available and are described in the main content and Appendix A.1. Detailed descriptions of the model architecture, training, and inference procedures are provided in Section 3.4 and Appendix A.1, and we explain and justify all hyperparameter choices in Appendix A.2 to ensure reproducibility.

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

---

**Algorithm 1** Training of RankFlow

---

**Input:** Pretrained language model PLM, pretrained structure encoder SE, training data $\mathcal{D} = \{(\boldsymbol{x}^{\text{mt}}, \boldsymbol{x}^{\text{wt}}, y)\}$, batch size $B$, learning rate $\eta$, weight $\lambda$, max epochs $E$, rank loss weight $\lambda_{\text{rank}}$, time sampler $t \sim \mathcal{U}(0,1)$
Initialize RankFlow parameters $\boldsymbol{\theta}$
$\boldsymbol{h}^{\text{wt}} = \text{PLM}(\boldsymbol{x}^{\text{wt}}), \quad \boldsymbol{s}^{\text{wt}} = \text{SE}(\boldsymbol{x}^{\text{wt}})$
**for** $epoch = 1$ **to** $E$ **do**
   **for** each batch $\{(\boldsymbol{x}_i^{\text{mt}}, \boldsymbol{x}^{\text{wt}}, y_i)\}_{i=1}^B$ in $\mathcal{D}$ **do**
      $\boldsymbol{h}_i^{\text{mt}} = \text{PLM}(\boldsymbol{x}_i^{\text{mt}})$,
      $F_i = \text{Encoder}([\boldsymbol{h}^{\text{wt}}; \boldsymbol{s}^{\text{wt}}])$;
      Sample $t_i \sim \mathcal{U}(0,1)$;
      Compute embeddings of mutated positions $\boldsymbol{c}_i(\boldsymbol{\mu}_i)$;
      $\boldsymbol{h}_i^{\text{mt}} = concat(\boldsymbol{h}_i^{\text{mt}}, \boldsymbol{c}_i(\boldsymbol{\mu}_i))$;
      Compute perturbed representation: $\boldsymbol{h}_{t_i} = \mu_{t_i}\boldsymbol{h}_i^{\text{mt}} + \sigma_{t_i}\epsilon$ where $\epsilon \sim \mathcal{N}(0, I)$;
      Compute target velocity $\boldsymbol{u}_{t_i}(\boldsymbol{h}_{t_i}|\boldsymbol{h}_i^{\text{mt}})$ using Eq. 3;
      Compute steering gate $\boldsymbol{g}$ using Eq. 16;
      $\boldsymbol{v}_{\boldsymbol{\theta}}^{t_i} = \text{FlowHead}(\boldsymbol{h}_{t_i}, t_i, [F_i; \boldsymbol{g}_i])$;
      Compute property-aware flow matching loss $\mathcal{L}_{\text{PFM}}$ using Eq. 8;
      Compute rank-wise loss $\mathcal{L}_{\text{RFlow}}$ using Eq. 10;
      $\mathcal{L} = \mathcal{L}_{\text{PFM}} + \mathcal{L}_{\text{RFlow}}$
      $\boldsymbol{\theta} \leftarrow \boldsymbol{\theta} - \eta\nabla_{\boldsymbol{\theta}}\mathcal{L}$
   **end for**
**end for**
**Return:** Trained RankFlow model parameters $\boldsymbol{\theta}$

---

# A  METHOD DETAILS

## A.1  DETAILS OF TRAINING

Algorithm 1 outlines the training procedure for RankFlow. The model parameters $\boldsymbol{\theta}$ are initialized, and the training proceeds for a specified number of epochs. In each epoch, the training data is processed in batches. For each assay, the wild-type sequence is encoded using the pretrained language model (PLM) and structure encoder (SE). Because each assay has a single wild-type sequence, its representation is computed once and reused across all batches.

## A.2  IMPLEMENTATION DETAILS

For the main results in Table 1, we use pretrained ESM2-650M as the sequence encoder and ESM-IF as the structure encoder, and the structures are predicted by AlphaFold2 (Jumper et al., 2021). For Table 2, which includes assays longer than 1,024 residues, we instead use pretrained ESMC-600M-2024-12 as the sequence encoder, while keeping ESM-IF as the structure encoder. Training uses grouped batches per assay and per protein. We use AdamW with learning rate $10^{-4}$, weight decay $10^{-2}$, batch size 256, and early stopping on validation set. Given the variability in the datasets, the number of training epochs is set to a maximum of 50. Training typically takes less than 1 hour on a single 80 GB A100 GPU. During inference, we use a fixed-step second-order explicit solver (Heun) over $t \in [1, 0]$ using the number of steps $N = 20$. From the endpoint, we read a scalar score by averaging mutant-vs-wild type logit changes over edited sites as in Eq. 9. This readout is invariant to the scale of logits and focuses on the mutations that drive the property change. For the mutational radius $r$, we use $r=2$ by default, and for extremely small assays, neighborhoods are sparse, $r$ will be increased to 3. During experiments, we fix $\lambda = 0.5$ after experimental validation. We use $\xi=0.3$ by default to form $\mathbb{S}^+$ and $\mathbb{S}^-$. If either split has fewer than 20 samples, we switch to a top-$k$ selection with $k = \min(20, \lfloor 0.3K \rfloor)$ per side. For very small assays ($K<80$), we disable PSG by setting all gates to 1. The hyper-parameter $\gamma$ is chosen from $\{0.5, 1.0\}$ based on dataset size. The temperature $\tau$ in Eq. 10 is set to 0.1. The weight $\lambda_{\text{rank}}$ in Eq. 10 is set to 0.5. $\beta$ in Eq. 7 is chosen from $\{1.0, 2.0, 5.0\}$ based on validation performance. For each assay, we compute Spearman's $\rho$ between predicted scores and ground-truth labels on the test set, and no additional calibration or post-processing is used.

Table 5: Comparison of different energy function designs on ProteinGym.

| Energy Function | Stability | Fitness | Expression | Binding | Activity |
|---|---|---|---|---|---|
| $\mathcal{E}_i(\boldsymbol{h}) = -\tilde{y}_i$ | 0.902 | 0.738 | 0.750 | 0.768 | 0.701 |
| $\mathcal{E}_i(\boldsymbol{h}) = -R(y_i)/n$ | 0.901 | 0.712 | 0.725 | 0.742 | 0.695 |
| $\mathcal{E}_i(\boldsymbol{h}) = -\left(\lambda\,\tilde{y}_i + (1-\lambda)\,\frac{\tilde{y}_i - \bar{\tilde{y}}_i}{\sqrt{s_i}}\right)$ (Ours) | **0.911** | **0.742** | **0.765** | **0.781** | **0.722** |

**Hyperparameter selection.**   We follow the commonly used training setup of flow-matching models and perform a coarse sweep on a held-out validation split over the three specific knobs that appear in Table 8: the objective, the energy-guidance weight, and the schedule. For the objective, the ablation shows that removing $RC^2$ or replacing it with another commonly used loss (e.g., kinetic reg.) degrades performance. For the energy-guidance weight $\lambda$, the value 0.5 yields the best average Spearman correlation, whereas both smaller and larger values lead to a noticeable drop in performance, so we fix $\lambda = 0.5$ for all experiments and recommend selecting $\lambda$ on a small validation set from a simple grid such as 0.25, 0.5, 1.0 in new applications. For the schedule, cosine scheduling performs slightly better than the linear variant, so we use it as the default and recommend it in the text.

The energy-guidance weight $\lambda$ is robust over a wide range, with $\lambda = 0.5$ performing best on average. Cosine scheduling consistently outperforms linear scheduling, so we use it in all reported results. Overall, RankFlow is robust to reasonable variations of these hyperparameters rather than relying on extensive benchmark-specific tuning.

(i) the objective, comparing variants without $RC^2$, with $RC^2$ only, and the full objective with kinetic regularization; (ii) the energy-guidance weight $\lambda \in \{0, 0.25, 0.5, 1\}$; and (iii) the schedule for $(\mu_t, \sigma_t)$, comparing linear and cosine scheduling. The ablation indicates that the full objective with $RC^2$ and kinetic regularization consistently outperforms the other variants, so we adopt this as the default. For the guidance weight, performance is relatively stable over the range $\lambda \in [0.25, 1]$, with $\lambda = 0.5$ achieving the best average Spearman correlation across the four ProteinGym properties; we therefore fix $\lambda = 0.5$ for all experiments and recommend $\lambda \in [0.25, 0.5]$ as a practical range. Cosine scheduling provides a small but consistent improvement over the linear schedule, so we use the cosine schedule in all reported results.

Overall, practitioners can reproduce our configuration by directly adopting these default settings, and the ablations in Table 8 show that RankFlow is robust to reasonable variations of these hyperparameters rather than relying on extensive benchmark-specific tuning.

## A.3   DESIGN DETAILS

**Design of Energy Function.**   We analyze the design of the energy function $\mathcal{E}_i(\boldsymbol{h}) = -\left(\lambda\,\tilde{y}_i + (1-\lambda)\,\frac{\tilde{y}_i - \bar{\tilde{y}}_i}{\sqrt{s_i}}\right)$ in Eq. 7 by comparing it with two alternatives: (1) a normalized score value $\mathcal{E}_i(\boldsymbol{h}) = -\tilde{y}_i$ without neighbor comparison, and (2) a rank-based score $\mathcal{E}_i(\boldsymbol{h}) = -R(y_i)/n$, where $R(y_i)$ is the rank of $y_i$ among all $n$ samples. The results are summarized in Table 5. The first alternative directly uses the normalized property value as the energy, which may be sensitive to outliers and does not consider the relative ranking among variants. The second alternative uses the rank as the energy, which captures relative ordering but loses fine-grained information about the magnitude of differences between variants. Our proposed energy function combines both absolute and relative information, leading to improved performance across all categories. This demonstrates that incorporating both the magnitude of the predicted property and its deviation from the local average provides a more informative signal for guiding the flow model in learning the fitness landscape.

**RankFlow Architecture Details.**   Our flow matching is based on a one-dimensional U-Net backbone (UNet1D) tailored for sequential data. The encoder consists of four residual blocks with interleaved linear attention and downsampling operations, reducing temporal resolution while enriching representations. A bottleneck stage applies two residual blocks with a full attention layer in between, capturing long-range dependencies. The decoder mirrors the encoder with four residual blocks and upsampling layers, combined with skip connections from the encoder. Temporal conditioning is incorporated through sinusoidal time embeddings processed by an MLP, while an additional MLP

Table 6: Model architecture details.

| Component | Details |
|---|---|
| Fusion Encoder | 2 MLPs with dim=1280, 1-layer Self-Attention, output dim=1280 |
| Flow Head | U-Net1D with 4 ResBlocks (down) + 2 ResBlocks (middle) + 4 ResBlocks (up), hidden dim=128, output dim=1280 |

handles auxiliary fitness embeddings. Finally, a residual block and $1\times$Conv1D projection layer yield the model's output. In total, UNet1D contains approximately 9.6M trainable parameters. The fusion encoder consists of two MLP layers with ReLU activations, followed by a self-attention layer to integrate sequence and structure features. The detailed architecture is summarized in Table 6.

# B EXPERIMENTAL DETAILS

## B.1 DETAILS OF BENCHMARK

ProteinGym is an extensive set of DMS assays, containing 217 datasets. Each dataset is categorized into one of five groups based on the measured property: Stability, Fitness, Expression, Binding, and Activity. For fair comparison, we implemented the Random cross-validation method as described in DePLM. Specifically, for each dataset, each mutation in the dataset is randomly assigned to one of five folds. The model is trained on four folds and evaluated on the remaining fold, and this process is repeated five times to ensure each fold serves as the test set once. The final performance is reported as the average across all five folds. For results reported in Table 2, which include all 217 assays, we follow ProteinGym's official evaluation protocols for the three splitting schemes: Random, Modulo, and Contiguous. In order to deal with

## B.2 EVALUATION METRIC

Following ProteinGym (Notin et al., 2023a), we quantify concordance with the experimental landscape using Spearman's $\rho$. For each protein assay, let the predicted scores be $\tilde{\boldsymbol{y}} = (\tilde{y}_i)_{i=1}^n$ and the experimental measurements be $\boldsymbol{y} = (y_i)_{i=1}^n$. Spearman's coefficient is the Pearson correlation between the ranked vectors:

$$\rho = \frac{\text{cov}\big(R(\tilde{\boldsymbol{y}}),\, R(\boldsymbol{y})\big)}{\sigma_{R(\tilde{\boldsymbol{y}})}\, \sigma_{R(\boldsymbol{y})}},$$

where $R(\cdot)$ is the vector of ranks (averaging ties). We compute $\rho$ per protein assay and report aggregate statistics across assays in the results.

## B.3 MULTIPLE MUTANTS PROPERTY PREDICTION

We further evaluate RankFlow on 15 protein assays with multiple mutants in ProteinGym. Figure 5 shows the Spearman's rank correlation between model predictions and experimental measurements for these assays. RankFlow consistently outperforms other baselines, demonstrating its effectiveness in capturing the complex interactions among multiple mutations. This highlights RankFlow's potential for guiding protein engineering efforts that involve combinatorial mutagenesis.

## B.4 CROSS-VALIDATION RESULTS

The error bars for the cross-validation setting are reported in Table 7. RankFlow surpasses Protein-NPT and DePLM across all categories while using fewer trainable parameters. Relative to DePLM, RankFlow improves average performance by 1.8% (Stability), 3.8% (Fitness), 2.2% (Expression), 1.7% (Binding), and 3.0% (Activity). It is also lighter (37.1M trainable parameters in RankFlow vs. 42.2M in DePLM) and faster to train (50 training epochs for RankFlow vs. 100 for DePLM). These results underscore RankFlow's effectiveness and parameter-efficient design.

In addition, we analyze the correlation of predictions from different methods. As shown in Figure 6, the predictions from RankFlow exhibit less correlation with those from DePLM and ESM-2 on the Fluorescence benchmark, indicating that RankFlow might capture complementary information.

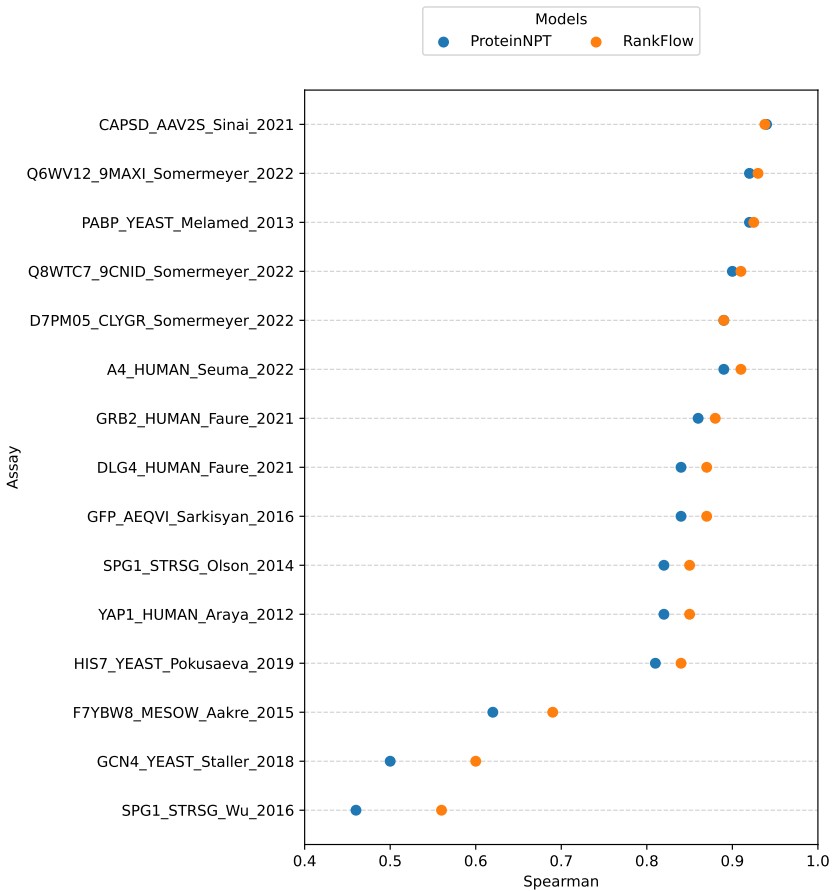

Figure 5: Spearman's rank correlation between model predictions and experimental measurements, for assays in ProteinGym with multiple mutants.

Table 7: Model performance on ProteinGym. We report mean±standard deviation performance under the Random scheme.

| Model | Stability | Fitness | Expression | Binding | Activity |
|---|---|---|---|---|---|
| ProteinNPT | 0.904±0.015 | 0.668±0.035 | 0.736±0.023 | 0.706±0.060 | 0.680±0.026 |
| DePLM | 0.897±0.013 | 0.707±0.027 | 0.742±0.027 | 0.764±0.041 | 0.693±0.024 |
| **RankFlow (Ours)** | **0.911±0.012** | **0.742±0.022** | **0.765±0.021** | **0.781±0.035** | **0.722±0.019** |

Additionally, the correlation coefficients on GB1 are significantly lower, suggesting that the models may be learning different aspects of the fitness landscape for this particular protein. We also found that ESM-2 and DePLM exhibit a stronger correlation between their predictions, which is consistent with their similar methodological basis. This suggests that RankFlow's unique architecture and training objectives enable it to learn distinct aspects of the protein fitness landscape, which may not be fully captured by other models. The moderate correlation also highlights the potential for ensemble approaches that combine predictions from multiple models to further enhance performance in protein fitness prediction tasks.

B.5    RESULT ANALYSIS OF $\beta$-LACT.

To understand how RankFlow achieves accurate predictions on hyperactive mutants, we extracted mutant embeddings of $\beta$-lact. from RankFlow, and visualized the latent space with t-SNE. As shown in Figure 7, mutants with high experimental fitness values (top quantile) are closer in the latent space,

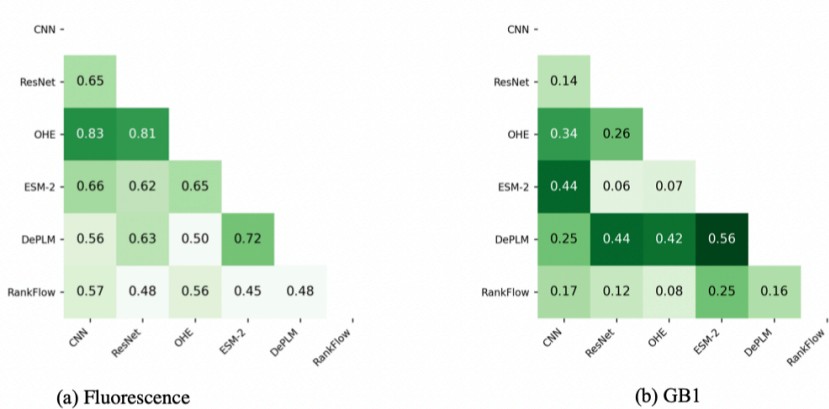

(a) Fluorescence   (b) GB1

Figure 6: The pair-wise Spearman's correlation coefficient of the predicted fitness values from different methods.

indicating that RankFlow effectively captures the underlying distribution of the fitness landscape. In contrast, mutants with low fitness values (bottom quantile) are more dispersed, suggesting that Rank-Flow can distinguish between high- and low-fitness variants. The t-SNE visualization also reveals that RankFlow organizes the mutant embeddings in a way that reflects their functional properties, which likely contributes to its strong performance in predicting protein fitness. This clustering of high-fitness mutants indicates that RankFlow has learned meaningful representations that correlate with experimental outcomes, enabling it to generalize well to unseen variants.

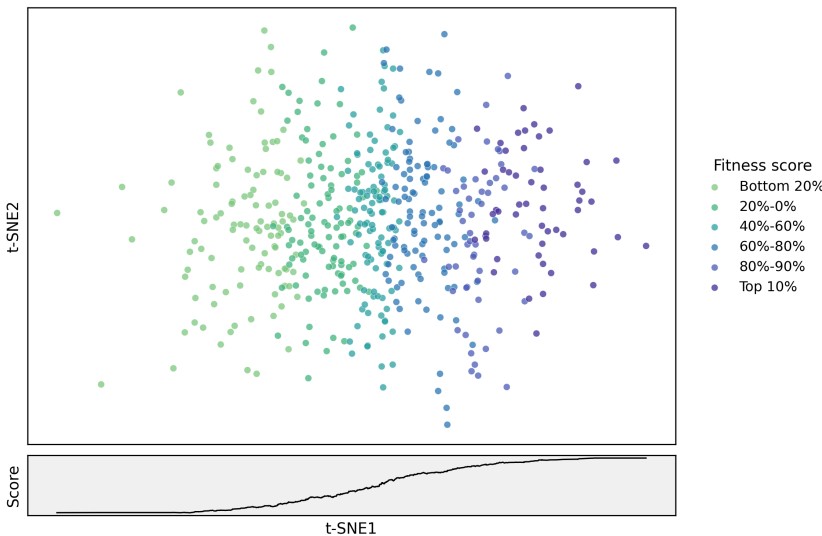

Figure 7: *t-SNE* visualization of $\beta$-lact. mutant embeddings from RankFlow. Points are colored by quantile bins of experimentally measured fitness; the bottom panel shows the ground-truth fitness trend.

## B.6 SUPPLEMENTARY ABLATION STUDY

Table 8 highlights the contribution of each design choice in RankFlow. Removing the rank-consistent loss ($\mathcal{L}_{\mathrm{RFlow}}$) leads to a noticeable degradation across all assay categories, especially in Fitness and Activity, confirming that ranking consistency is crucial under data-scarce conditions. Using $RC^2$ alone performs poorly, indicating that flow matching remains necessary to anchor the transport map. Adding a simple kinetic regularizer recovers only part of the lost performance, underscoring the importance of our rank-aware objective. For the energy design, both extremes ($\lambda$=1

Table 8: Ablations of RankFlow on ProteinGym.

| Model Variant | Stability | Fitness | Expression | Binding | Activity |
|---|---|---|---|---|---|
| Full RankFlow | **0.911** | **0.742** | **0.765** | **0.781** | **0.722** |
| *Objective Loss* | | | | | |
| w/o $\mathcal{L}_{\text{RFlow}}$ | 0.885 | 0.702 | 0.736 | 0.744 | 0.680 |
| $\mathcal{L}_{\text{RFlow}}$ only | 0.881 | 0.624 | 0.676 | 0.622 | 0.613 |
| $\mathcal{L}_{\text{PFM}}$ + kinetic reg. | 0.892 | 0.714 | 0.745 | 0.755 | 0.695 |
| $\mathcal{L}_{\text{PFM}}$ + pairwise hinge | 0.889 | 0.709 | 0.740 | 0.751 | 0.690 |
| *Energy guidance* | | | | | |
| $\lambda = 1$ | 0.902 | 0.738 | 0.750 | 0.768 | 0.701 |
| $\lambda = 0$ | 0.896 | 0.718 | 0.744 | 0.756 | 0.699 |
| $\lambda=0.25$ | 0.900 | 0.719 | 0.748 | 0.759 | 0.702 |
| $\lambda=0.50$ | 0.911 | 0.742 | 0.765 | 0.781 | 0.722 |
| *Scheduling* | | | | | |
| Linear $(\mu_t, \sigma_t)$ | 0.908 | 0.736 | 0.762 | 0.772 | 0.718 |
| Cosine $(\mu_t, \sigma_t)$ | 0.911 | 0.742 | 0.765 | 0.781 | 0.722 |

Table 9: Comparison of computational costs of ProteinNPT and DePLM.

| Method | Training (MACs) | Inference (MACs) | Parameter (#) |
|---|---|---|---|
| ProteinNPT | $58.63\text{M} = 11724.82 \times 5001$ | 11.7K | $100\text{M} + 119\text{M}$ |
| DePLM | $\begin{aligned}\mathbf{9.16K} &= 180.56 + 77.55 \\ &+89.05 \times 100\end{aligned}$ | $347.16 = 180.56 + 77.55 + 89.05$ | $792\text{M} + 42.2\text{M}$ |
| **RankFlow (Ours)** | $\begin{aligned}897.1\text{K} &= 179.25 \times 5001 \\ &+77.55 + 19.34 \times 30\end{aligned}$ | **19.34** | $792\text{M} \ \mathbf{+37.1M}$ |

using only global magnitude or $\lambda=0$ using only local deviation) underperform, while intermediate values achieve stronger results; the best trade-off is obtained at $\lambda=0.5$, showing that global fitness and local substitution deviation are complementary signals. Finally, cosine scheduling consistently outperforms a linear schedule, particularly in Fitness and Binding, demonstrating that smoother variance growth improves the stability of the learned flow. Overall, the ablations validate that each proposed component: $\mathcal{L}_{\text{RFlow}}$, balanced energy guidance, and cosine scheduling contributes to Rank-Flow's state-of-the-art performance.

### B.7 COMPUTATIONAL COSTS

Following Wang et al. (2024), we use A4GRB6_PSEAI as an example to compare the computational costs of different models. The sequence length is 267, and the dataset size is 5001. We report the number of trainable parameters, peak GPU memory usage, and training time in Table 9. For both DePLM and RankFlow, the sequence encoder requires 180.56G MACs and 179.25G MACs, respectively, as RankFlow only requires internal representation from the sequence encoder not the final predicted logits. The structure encoder requires 77.55G MACs for both models. Before training a model on a given assay, both ProteinNPT and RankFlow compute and persist to disk the sequence embeddings for all mutated proteins in that assay. During training, we load from disk the embeddings corresponding to each batch. RankFlow trains with higher MACs than DePLM (897.1K vs. 9.16K) as it learns a property-aligned velocity field for all mutants. However, for this assay, RankFlow lowers total inference cost by 94.4% compared to DePLM, making it more practical for large-scale screening. RankFlow also uses fewer trainable parameters (37.1M vs. 42.2M) by employing a lightweight flow architecture instead of a large transformer decoder, reducing memory footprint and speeding up training. Overall, RankFlow achieves a favorable balance of training efficiency and inference speed compared to ProteinNPT and DePLM.

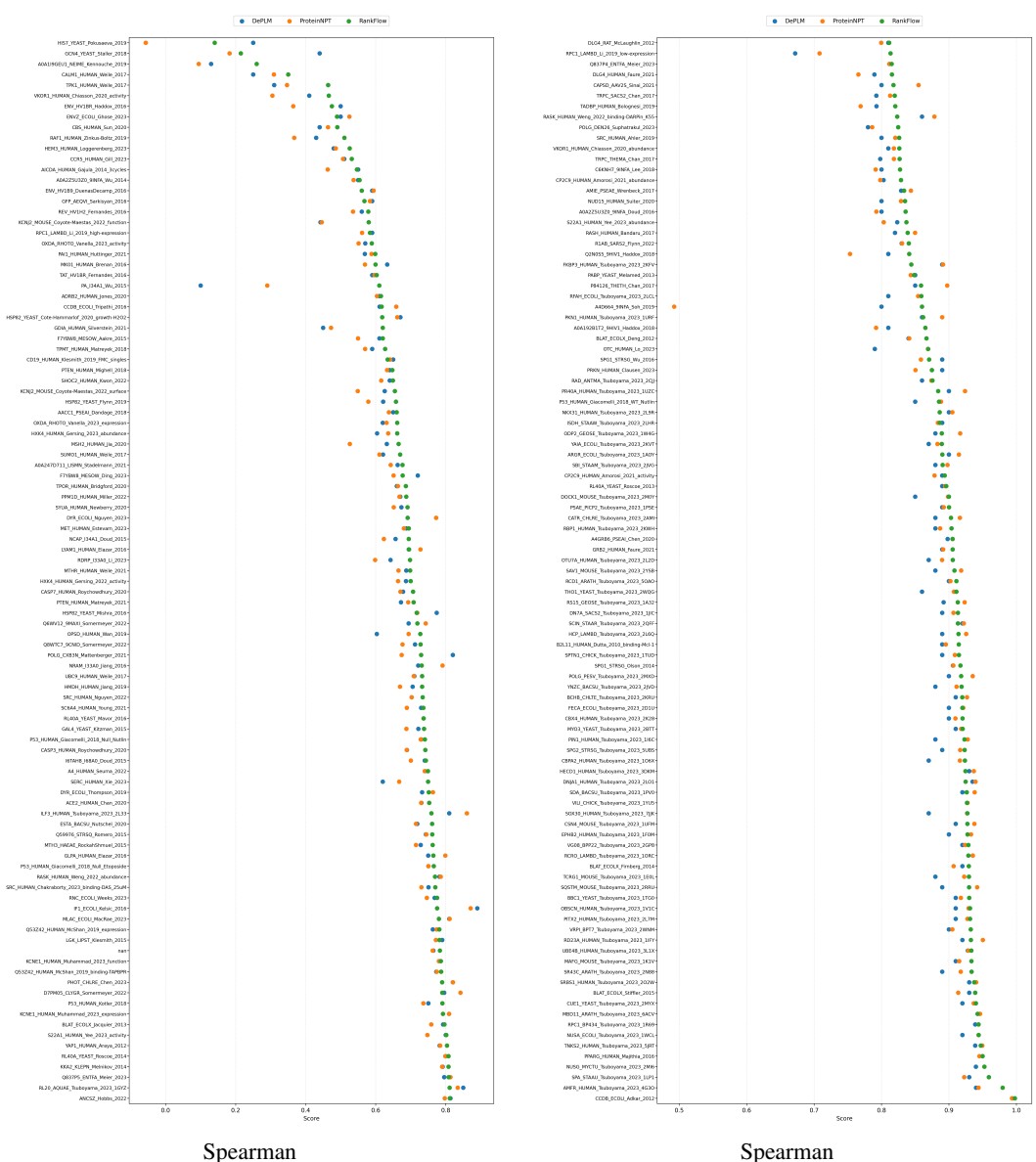

Figure 8: Results under the random cross-validation scheme on ProteinGym. We report the DMS-level performance of DePLM, ProteinNPT, and our RankFlow.

## C BACKGROUND

### C.1 CONDITIONAL FLOW MATCHING

Continuous Normalizing Flows (CNFs) (Chen et al., 2018) consider the dynamics of a probability density via a path $p : [0, 1] \times \mathbb{R}^d \rightarrow \mathbb{R}_{\geq 0}$ that transports between data distribution $p_0$ and an initial distribution $p_1$. The flow is governed by a vector field $\boldsymbol{v} : [0, 1] \times \mathbb{R}^d \rightarrow \mathbb{R}^d$ through the continuity equation:

$$\partial_t p_t(\boldsymbol{h}) + \nabla \cdot (p_t(\boldsymbol{h})\boldsymbol{v}(\boldsymbol{h}, t)) = 0, \quad p_{t=0} = p_0, \quad p_{t=1} = p_1. \tag{17}$$

The objective is to learn the time-dependent vector field $\boldsymbol{v}$ describing the velocity of a particle at position $\boldsymbol{h}$. In practice, given the conditional distribution $p_t(\boldsymbol{h} \,|\, \boldsymbol{h}_0)$, which is usually modeled as a Gaussian path, i.e., $p_t(\boldsymbol{h} \,|\, \boldsymbol{h}_0) = \mathcal{N}(\mu_t \boldsymbol{h}_0, \sigma^2 \boldsymbol{I})$, where $\mu_t, \sigma_t$ is Gaussian scheduler. The vector field $\boldsymbol{v}$ is learned by minimizing the flow matching objective (Lipman et al., 2022):

$$\mathcal{L}_{\text{FM}}(\boldsymbol{\theta}) = \mathbb{E}_{t \sim \mathcal{U}(0,1), \boldsymbol{h}_0 \sim p_0, \boldsymbol{h}_1 \sim p_1, \boldsymbol{h}_t \sim p_t(\cdot | \boldsymbol{h}_0)} \big[ \| \boldsymbol{v}_{\boldsymbol{\theta}}(\boldsymbol{h}_t, t) - \boldsymbol{u}_t(\boldsymbol{h}_t | \boldsymbol{h}_0) \|_2^2 \big], \tag{18}$$

where $\boldsymbol{h}_0$ follows the data distribution $p_0$.

### C.2 PROTEIN LANGUAGE MODELS FOR MUTATIONAL EFFECT PREDICTION

Protein Language Models (PLMs) are deep learning models trained on large corpora of protein sequences using self-supervised learning objectives, such as masked language modeling (MLM) or autoregressive modeling. These models learn to capture the statistical patterns and dependencies in protein sequences, enabling them to generate meaningful representations that can be used for various downstream tasks, including mutational effect prediction. Protein language models trained using the masked language modeling objective, such as ESM-2 (Lin et al., 2023), can be used to predict the effects of mutations on protein function. These models are trained to predict masked amino acids in a sequence based on the surrounding context, allowing them to learn rich representations that capture the relationships between different amino acids and their roles in protein structure and function. Specifically, given a protein sequence $\boldsymbol{x} = (x_1, x_2, \ldots, x_N)$, where $N$ is the length of the sequence and $x_i$ represents the amino acid at position $i$, a PLM trained with MLM learns to predict the masked amino acid $x_m$ given the context of the other amino acids in the sequence. The model outputs a probability distribution over the 20 standard amino acids for each position in the sequence. For a mutant sequence $\boldsymbol{x}^{\text{mt}}$ and its corresponding wild type sequence $\boldsymbol{x}^{\text{wt}}$, we can compute the log-odds score for each mutation at position $m$ as follows:

$$\sum_{m \in \boldsymbol{\mu}} \big( \log P(x_m = \boldsymbol{x}_m^{\text{mt}} | \boldsymbol{x}_{\backslash m}^{\text{mt}}) - \log P(x_m = \boldsymbol{x}_m^{\text{wt}} | \boldsymbol{x}_{\backslash m}^{\text{mt}}) \big), \tag{19}$$

where $\boldsymbol{\mu}$ is the set of mutated positions, and $\boldsymbol{x}_{\backslash m}^{\text{mt}}$ denotes the mutant sequence with the amino acid at position $m$ masked. This log-odds score reflects the model's confidence in the mutant amino acid relative to the wild type amino acid, providing an estimate of the mutation's effect on protein function (Meier et al., 2021). By leveraging the learned representations from PLMs, we can effectively predict the impact of mutations on protein properties, aiding in tasks such as protein engineering and understanding disease-related mutations.

## D LIMITATION AND FUTURE WORK

While RankFlow demonstrates strong performance in protein fitness prediction, there are limitations and areas for future work. First, the reliance on pretrained language models means that the quality of the embeddings is contingent on the diversity and representativeness of the training data used for these models. Future work could explore fine-tuning or adapting PLMs specifically for protein engineering tasks to enhance their relevance. Second, while RankFlow effectively integrates structural information through ESM-IF, it may not fully capture dynamic aspects of variant structures that are crucial for function. Incorporating more sophisticated structural representations or dynamic simulations could further improve performance. Finally, while RankFlow is efficient compared to some baselines, further optimizations in model architecture and training procedures could make it more accessible for large-scale applications in protein engineering. For example, exploring lightweight architectures or distillation techniques for extracting protein representations could reduce computational costs without sacrificing accuracy.

# E    THE USE OF LARGE LANGUAGE MODELS (LLMS)

Our paper uses LLMs to polish the writing. We did not use LLMs to generate any scientific content, including the main ideas, algorithms, or experimental results. All scientific content was developed and verified by the authors.

