# OpenReview forum: "RankFlow: Property-aware Transport for Protein Optimization"
_ICLR.cc/2026/Conference — ICLR 2026 Poster_

### Official Review · Reviewer_XW2e · 2025-10-26

**Soundness:** 2
**Presentation:** 2
**Contribution:** 2
**Rating:** 2
**Confidence:** 3

**Summary:**

The paper presents RankFlow, a method for improving protein fitness prediction using flow matching. The main contribution is reformulating fitness prediction as a generative task rather than a regression problem. While the proposed approach is potentially interesting, the paper requires major revisions, particularly to improve clarity in the overall motivation and experimental setup.

**Strengths:**

- The idea of applying flow matching to protein fitness prediction appears novel.
- The proposed components, such as the Rank-Consistent Conditional Flow Loss and the Property-Guided Steering Gate, are original and well-motivated.

**Weaknesses:**

Major comments

- Although the method’s technical details are described, the overall intuition and practical usage remain unclear. The motivation behind reformulating fitness prediction as a generative problem (rather than regression) is not well explained. A conceptual figure illustrating this idea would be helpful (see, for example, Fig. 2 in [1]). In addition, several typos make the method difficult to follow (see minor comments).
- The experimental setup is insufficiently described:
  - Evaluation metrics are not mentioned in the main text.
  - The values reported in Table 1 and Table 2 do not match those on the official ProteinGym website [2], and no explanation is provided. For example, ProteinNPT achieves a “Stability” score of 0.904 in the paper, while [2] reports 0.776.
- The evaluation mixes unsupervised models (which do not use any fitness data, e.g., ESM-2, SaProt) with supervised models (e.g., ProteinNPT) without justification or explanation. It is unclear what training data RankFlow uses. The parameter count comparison in the table appears inconsistent. How can RankFlow have fewer parameters than ESM-2 if it uses ESM-2 as its backbone?

Minor comments

- [Line 152] x^{mt} is likely a typo; it should be x^{wt}.
- [Line 257] R_{\tau} is mentioned but never explained.
- [Line 273 and Eq. 13] The notation for S^+ and S^- is inconsistent.
- [Eq. 16] g_i is introduced here but its role is not clarified in the rest of the technical explanation of the method.
- [Line 372] The text claims: “As shown in Fig. 3, variants predicted to have high fitness are at solvent-exposed positions and are located away from the active site, which is in line with established biological knowledge (Notin et al., 2022a).” However, Fig. 3 does not demonstrate this.

[1] Corso et al., 2023. DiffDock: Diffusion Steps, Twists, and Turns for Molecular Docking.https://arxiv.org/abs/2210.01776

[2] https://proteingym.org/

**Questions:**

1. Is performance difference between RankFlow and DePLM statistically significant?

---

> ### Author Response · Authors · 2025-11-21
>
> Thank you very much for your detailed review and for explicitly highlighting our idea as novel, original and well-motivated.
>
> **Weakness** **- The overall intuition, practical usage, comparison with regression head, and a conceptual figure**
>
> The original Introduction aimed to provide an intuitive description of RankFlow as a property-aware conditional flow that refines wild-type-conditioned PLM representations using an energy function, a rank-consistent objective, and a property-guided steering gate. However, we acknowledge that the comparison to standard regression heads was not sufficiently emphasized. **In summary, an energy-guided generative flow provides a more structured and generalizable way to adapt PLM representations to fitness than a regression head.** We acknowledge that attaching a regression head to a pretrained PLM and fine-tuning on a single DMS assay can yield strong performance on that assay, but it is free to rescale and distort the PLM embeddings to match a particular label range and pattern of that assay, which tends to introduce dataset-specific bias and overwrite the generalization capabilities of pretrained PLMs. In contrast, our design keeps the PLM as a base sequence/structure prior and learns a conditional flow whose evolution is guided by an energy function constructed from fitness scores. The flow is trained to reshape wild-type-conditioned mutant representations into a fitness-aligned distribution, and the rank-consistent loss focuses on getting the ordering of mutants right rather than matching their absolute values. In other words, the PLM still encodes which mutants are plausible, while our energy function and rank-consistent objective only reshape this prior so that high-fitness mutants are consistently ranked above low-fitness ones, without arbitrary changes to the embedding space. This results in a more structured form of adaptation than using a regression head and helps preserve the PLM’s generalization ability. As shown in our generalizability experiments (Section 4.2, now Section 4.3), in the cross-assay setting with 40 training assays and one held-out test assay, both fine-tuned ESM2 (Spearman ρ = 0.364) and fine-tuned SaProt (ρ = 0.505) with regression heads underperform, whereas RankFlow achieves the highest Spearman correlation (ρ = 0.571), indicating superior generalization. We have revised the paper to include these discussions (Lines 80-86).
>
> We have also revised Figure 1 as a conceptual figure to clarify both the intuition and the practical usage of RankFlow. Figure 1 (a) now illustrates that a regression head on PLM embeddings can overfit when fine-tuned on a single assay, mapping some truly high-fitness mutants to low scores and vice versa. RankFlow instead reshapes wild-type-conditioned mutant representations into a fitness-aligned distribution, enforcing a property-aware landscape. Figure 1 (b) connects this intuition to practice by depicting our cross-assay generalization protocol, where RankFlow achieves higher Spearman correlation on a held-out assay than regression-based fine-tuning (ESM2/SaProt). We also carefully corrected the typos highlighted in the minor comments to improve readability.
>
> **Weakness** **- Evaluation metrics not in main text**
>
> We now describe the evaluation protocol in the main text (Lines 360-362) and not only in the Appendix: Following ProteinGym, we use the Spearman's rank correlation coefficient between predicted and true fitness values as the evaluation metric.
>
> **Weakness** **- Discrepancy with the scores on the official ProteinGym website**
>
> As clarified in Appendix B.1, results of Table 1 follow the DePLM’s evaluation setup rather than the full ProteinGym benchmark. Specifically, we exclude assays whose wild-type sequence length exceeds 1,024 residues due to the context-length limitation of pretrained ESM2, yielding 201 DMS datasets. Under this protocol, ProteinNPT attains a stability score of 0.904, higher than the 0.776 reported on the ProteinGym website. For Table 2, as described in our generalizability experiments (Section 4.2, now Section 4.3), we use a different training/evaluation procedure: for a given test dataset, we randomly select up to 40 datasets from the same functional category as the training dataset, while enforcing sequence similarity between training and test sequences below 50% to prevent data leakage.

---

> > ### Author Response · Authors · 2025-11-21
> >
> > **Weakness** **- Evaluation mixes unsupervised models with supervised models**
> >
> > In Table 1, all methods are evaluated in a supervised setting: all models are either trained from scratch (e.g., CNN, Transformer, ProteinNPT, DePLM, RankFlow) or fine-tuned (e.g., ESM2, ESM-1v) on ProteinGym fitness labels. We have updated the table’s caption to state this explicitly. In Table 2 (now Table 4), we now follow the reviewer’s suggestion and group methods into two groups: zero-shot models that do not use any DMS fitness labels (e.g., ESM2, ProtSSN, ProteinMPNN) and supervised models that are trained/fine-tuned on ProteinGym assays (e.g., ESM2(FT), ProtSSN (FT), DePLM, RankFlow).
> >
> > **Weakness** **- Clarification of training data**
> >
> > The training protocol for RankFlow is already specified in the submitted manuscript. For Table 1, Appendix B.1 describes that, for each ProteinGym dataset, we use a five-fold random cross-validation scheme: each mutation is randomly assigned to one of five folds, RankFlow is trained on four folds and evaluated on the remaining fold, and the final performance is reported as the average across all five folds. To make this clearer at a glance, we have also revised the caption of Table 1 to read: “Spearman performance of supervised methods on $\beta$-lact., GB1, Fluo., and ProteinGym under the Random scheme.”
> >
> > For Table 2 (now Table 4), Section 4.2 already specifies that for each test dataset, we randomly select up to 40 datasets from the same ProteinGym category as training data, while ensuring that sequence similarity between training and test datasets remains below 50% to avoid data leakage.
> >
> > **Weakness** **- Parameter counts**
> >
> > Our intention in Table 2 was to report the number of trainable parameters, not the full backbone size. In the original submission, we mistakenly included parameter counts for zero-shot methods, which have no task-specific trainable parameters. We now include the “Trainable parameters” column, which is only for supervised or fine-tuned methods (ESM2(FT), SaProt(FT), ProteinNPT, DePLM, RankFlow). Although RankFlow uses ESM-2 as a backbone to extract PLM embeddings, this backbone is kept **frozen** and only a conditional flow-matching model is trained, so its reported parameter count is smaller than that of fully fine-tuned ESM-2.
> >
> > **Question** **- Statistical significance of RankFlow vs. DePLM**
> >
> > We found that the gains are especially pronounced on challenging assays, where DePLM achieves relatively low correlations. For example, on PA_I34A1_Wu_2015, RankFlow improves Spearman from 0.10 to 0.41; on A0A1I9GEU1_NEIME_Kennouche_2019, the gain is from 0.13 to 0.26; and on CALM1_HUMAN_Weile_2017, from 0.25 to 0.35.
> >
> > **Minor comment** **- Typo of $x^{\text{mt}}$ and notation for S^+ and S^-**
> >
> > We have correctified to $x^{\text{wt}}$ and updated the notations for S^+ and S^-.
> >
> > **Minor comment** **- R\_{\tau} is mentioned but never explained**
> >
> > In the original submission, we did define “$R\_{\tau}$ is a soft-ranking operator with temperature $\tau$ > 0”, which is brief. In the revision, we have clarified that $R\_{\tau}$ is a standard differentiable ranking operator used in prior work on differentiable sorting and ranking (e.g., “Differentiable Ranks and Sorting using Optimal Transport”, NeurIPS 2019).
> >
> > **Minor comment** **- Role of $\boldsymbol{g}\_i$**
> >
> > $\boldsymbol{g}\_i$ In Eq. 16 denotes the steering gate for mutant $i$, then it is referred to as the “steering gate $\boldsymbol{g}$” in Section 3.4 (“…the condition $\boldsymbol{C}$ which includes the fused representation $\boldsymbol{F}$ and the steering gate $\boldsymbol{g}$…”, Line 331), where we omitted the index $i$ for brevity. In the revised version, we have clarified this connection explicitly and replace “the steering gate $\boldsymbol{g}$” with “the steering gate $\boldsymbol{g}\_i$ obtained from Eq. 16”.
> >
> > **Minor comment** **- Content of Figure 3**
> >
> > We have revised Figure 3 to clearly indicate the solvent-exposed area (Residues 117-120, 121-122) and the active site (Residue 58). As shown in the updated figure, residues F115, D118, and E122, which are predicted to have high fitness, are indeed surface-exposed and located far from the active site.

---

> > > ### Comment · Reviewer_XW2e · 2025-11-25
> > >
> > > I appreciate the author's comments.
> > >
> > > > *Question - Statistical significance of RankFlow vs. DePLM*
> > >
> > > > We found that the gains are especially pronounced on challenging assays, where DePLM achieves relatively low correlations. For example, on PA_I34A1_Wu_2015, RankFlow improves Spearman from 0.10 to 0.41; on A0A1I9GEU1_NEIME_Kennouche_2019, the gain is from 0.13 to 0.26; and on CALM1_HUMAN_Weile_2017, from 0.25 to 0.35.
> > >
> > > Could the authors please provide a result of a statistical test? For example, paired t-test or other suitable test?

---

> > > > ### Author Response · Authors · 2025-11-26
> > > >
> > > > We thank the reviewer for this follow-up question and agree that a formal statistical test would be desirable. Unfortunately, the original DePLM paper does not release per-assay ProteinGym scores, only aggregate means and a summary comparison figure to ProteinNPT are reported, from which exact Spearman values cannot be recovered. A paired test (e.g., paired t-test or Wilcoxon signed-rank test) requires matched per-assay scores for both methods, so it is not possible to compute such a test using the official DePLM results.
> > > >
> > > > We did attempt to re-run DePLM and compute per-assay correlations, but these values seem to differ from those reported in the paper, likely due to implementation details. Instead, we report effect sizes for several challenging assays where DePLM's published performance is low and RankFlow provides large, practically meaningful gains (e.g., PA_I34A1_Wu_2015: 0.10→0.41; A0A1I9GEU1_NEIME_Kennouche_2019: 0.13→0.26; CALM1_HUMAN_Weile_2017: 0.25→0.35).

---

> > > > ### Author Response · Authors · 2025-11-27
> > > >
> > > > Dear Reviewer XW2e,
> > > >
> > > > We hope this message finds you well, and we sincerely appreciate the time and effort you have devoted to reviewing our work. As the discussion period is nearing its end, we wanted to kindly ask whether you have any additional comments or concerns that we could help clarify. We would be happy to provide any further details that might assist in your evaluation.
> > > >
> > > > Thank you again for your valuable feedback.
> > > >
> > > > Best regards,
> > > > Authors

---

> ### Author Response · Authors · 2025-11-28
>
> We followed the reviewer’s suggestion and evaluated the performance difference between RankFlow and DePLM using both a paired $t$-test and a Wilcoxon signed-rank test on per-assay Spearman correlations. Both tests show that the difference in per-assay Spearman correlation between RankFlow and DePLM is statistically significant.
>
> Concretely, we ran DePLM using the official code and all hyperparameters reported in their paper to reproduce the DMS-level per-assay scores. Let $\Delta\_i = \rho^{\text{DePLM}}\_i - \rho^{\text{RankFlow}}\_i$ denote the per-assay differences. Formal normality tests (Shapiro–Wilk and D’Agostino’s $K^2$) reject an exact Gaussian model for $\Delta_i$ at our sample size ($N = 201$), while visual inspection of the distribution suggests it is approximately normal with slight skewness and heavy tails. For this reason, we report both a parametric (paired $t$-test) and a non-parametric (Wilcoxon signed-rank) test.
>
> Across the $N = 201$ ProteinGym substitution assays ($df = 200$), DePLM and RankFlow obtain mean Spearman correlations of $\overline{\rho}^{\text{DePLM}} = 0.761$ and $\overline{\rho}^{\text{RankFlow}} = 0.784$, respectively, corresponding to a mean per-assay difference of $\Delta_{\text{mean}} = -0.022$ (median $\Delta_{\text{median}} = -0.019$) in the $\rho^{\text{DePLM}} - \rho^{\text{RankFlow}}$ convention. The paired $t$-test yields $t = -5.82$, $p = 2.27 \times 10^{-8}$ and a 95\% confidence interval for the mean difference of $[-0.0297,-0.0147]$, indicating a clear and statistically significant advantage for RankFlow.
>
> The Wilcoxon signed-rank test on the same differences is also highly significant ($W = 3270.0$, $p =  7.9 \times 10^{-17}$), confirming that this conclusion does not rely on the Gaussian assumption. Taken together, these results robustly highlight the superior performance of RankFlow, and we hope this analysis fully addresses the reviewer’s concern about the statistical significance of the performance gap between RankFlow and DePLM. We also provide a detailed per-assay comparison in Figure 8 of the revised manuscript.
>
> **Paired $t$-test on per-assay Spearman correlations (N = 201) ProteinGym substitution assays**
>
> | Quantity                                            | Value                                  |
> |-----------------------------------------------------|----------------------------------------|
> | Number of assays $N$                            | 201                                    |
> | Degrees of freedom $df$                      | 200                                    |
> | Mean Spearman, DePLM $\overline{\rho}^{\text{DePLM}}$     | 0.761                                  |
> | Mean Spearman, RankFlow $\overline{\rho}^{\text{RankFlow}}$ | 0.784                                  |
> | Mean difference $\Delta\_{\text{mean}} = \rho^{\text{DePLM}} - \rho^{\text{RankFlow}}$ | -0.022                             |
> | 95% CI for mean difference                          | [-0.0297,-0.0147]                |
> | Paired $t$-statistic                              | -5.82                              |
> | $p$-value                                         | $2.27 \times 10^{-8}$                 |

---

### Official Review · Reviewer_3v7D · 2025-10-31

**Soundness:** 3
**Presentation:** 3
**Contribution:** 2
**Rating:** 6
**Confidence:** 4

**Summary:**

The authors propose a flow-matching-based framework for protein fitness prediction. For that, they design a new energy function that is combined with representations from pLMs. They also add a gate mechanism to leverage mutational information from a dataset to bias mutations in the framework. Results show that the proposed method achieves state-of-the-art performance for the ProteinGym benchmark and three more tasks. Ablation studies show that the proposed components, especially the proposed energy function, are crucial to achieve good performance.

**Strengths:**

1. The authors propose a conditional flow framework that learns property-aligned embeddings that improve fitness prediction, especially for cases in which mutants have multiple mutations.
2. A new energy function (Eq. 7) and loss are proposed for protein fitness prediction.
3. The proposed RankFlow method achieves state-of-the-art results on important benchmarks such as ProteinGym.

**Weaknesses:**

1. Hyperparameters that should be tuned to achieve state-of-the-art performance are discussed only in the Appendix.
2. The implementation of the flow-based method is not clear when following the information in Section 3.2.
3. Additional analysis of the method for proteins without structure available (without the multimodal fusion encoder) seems needed.
4. Code is not available.

**Questions:**

Overall, the paper contributes a new flow-based framework for protein fitness prediction that could be a contribution to epistasis modeling. My initial recommendation is borderline acceptance, but I would like to discuss with the authors the following comments.

Comments:

1. The proposed method uses structure as input. For the datasets experimented, are all structures experimental or some are predicted? A clarification regarding which benchmark methods use sequence, structure, and sequence-structure seems needed.
2. Many hyperparameters are included in Section 3, and additional information is given only in the Appendix. As these seem to directly influence the state-of-the-art performance of the proposed method, more discussion about how to choose these hyperparameters is needed.
3. For the gate, how much assay data is needed for this term to be effective? Does the nature of assay data influence this term?
4. (lines 315-316) Can the authors elaborate on what it means that the learnable embeddings are applied only at the mutated positions?
5. Additional clarification regarding the modeling of the energy function in Eq. 7 seems needed. The ablations in the Appendix show that this term is crucial for performance.

Minor Comments (that did not impact the score):

1. The use of \citet and \citep is wrong throughout the manuscript. Almost all references are missing parentheses.
2. The acronym of pLMs is defined twice in text.
3. Need consistency in the use of hyphens, e.g., “deep-mutational-scanning”; and when using a comma for numbers.
4. Figure 4 should be improved as it is very hard to read the caption in the printed version.
5. Typo: “reproducibility”

---

> ### Author Response · Authors · 2025-11-21
>
> Thank you very much for your thorough review and all valuable and constructive comments.
>
> **Weakness** **1 - Discussion of hyperparameters in main context**
>
> In the revised Section 4.1, we have explained that we perform coarse validation and then fix a single configuration for all ProteinGym datasets. Section 4.1 now summarizes main hyperparameters, with full details in Appendix B.6. Specifically: *Loss composition.* We start from the base PFM loss $\mathcal{L}\_{\text{PFM}}$ and compare: (i) $\mathcal{L}\_{\text{PFM}}$ only (w/o $\mathcal{L}\_{\text{RFlow}}$), and (ii) replacing $\mathcal{L}\_{\text{RFlow}}$ with kinetic or pairwise-hinge regularisation. Table 8 (top block) shows that all variants are uniformly worse compared to our full loss term.
> *$\lambda$ in the energy function.* We sweep $\lambda \in$ {0, 0.25, 0.5, 1}. Performance varies smoothly and $\lambda$ = 0.5 gives the best or tied-best Spearman across properties (Table 8, middle).
> *Scheduling.* We compare linear vs cosine schedules for ($\mu_t$,$\sigma_t$). Cosine is consistently but modestly better (Table 8, bottom).
>
> **Weakness** **2 - Flow-based implementation unclear**
>
> In the revision, we have added one paragraph (Lines 330-341) at the end of Section 3.4 to explicitly describe how RankFlow is implemented in practice. Section 3.2 is intended to present the energy-guided flow formulation, and Sections 3.3-3.4 introduce the architectural components; placing this implementation summary as the final paragraph of Section 3.4 provides a clear bridge between the theoretical description and the concrete algorithm.
> Concretely, for each mutant $x_i^{\mathrm{mt}}$, we first obtain its representation $\boldsymbol{h}\_i^0$ from the frozen PLM, together with the fused sequence-structure representation $\boldsymbol{F}$ and steering gate $\boldsymbol{g}\_i$; these are collected into the condition $\boldsymbol{C}\_i = $[$\boldsymbol{F}$, $\boldsymbol{g}\_i$]. We then sample a time $t \sim \mathcal{U}(0,1)$, construct a noisy state $\boldsymbol{h}\_{t} = \mu_t \boldsymbol{h}^0 + \sigma\_t \varepsilon$ with fixed scheduler $(\mu_t,\sigma_t)$ and $\varepsilon \sim \mathcal{N}(0,I)$, and compute the target velocity $\boldsymbol{u}\_t(\boldsymbol{h}\_{t}|\boldsymbol{h}^0)$ in closed form according to Eq. 3, where we omit the index $i$ for brevity. The flow head takes $(\boldsymbol{h}\_{t}, \boldsymbol{C})$ as input and predicts a velocity $\boldsymbol{v}\_t(\boldsymbol{h}\_t;\theta)$. The parameters $\theta$ are optimized by minimizing the energy-weighted flow-matching loss $\mathcal{L}\_{\text{PFM}}$ in Eq. 8 together with the rank-consistent loss in Eq. 10. Algorithm 1 in Appendix A.1 summarizes the overall training procedure. At inference time, we fix $\boldsymbol{C}$ for a given assay and mutant, integrate the learned vector field from $t=0$ to $t=1$ using a fixed-step Heun solver with $N$ steps (Appendix A.2), and map the final representation to a scalar fitness score.
>
> **Weakness 3** **- Analysis of the method without structure available**
>
> We have added an ablation where we disable the multimodal fusion encoder and the structure branch, and apply RankFlow in a sequence-only setting (“RankFlow w/o Structure info” in Figure 4(a), Section 4.3). We observe a moderate performance drop when structure is removed, especially on label-sparse ProteinGym assays, notably within the Fitness (0.715 vs. 0.742) and Activity (0.695 vs. 0.722) categories. However, it still achieves better performance than strong sequence-only baselines such as ESM2(FT) and SaProt(FT). This analysis shows that RankFlow can still be applied meaningfully to proteins without available structures, while the multimodal fusion encoder provides additional gains when structure is present.
>
> **Weakness 4** **- Code is not available**
>
> We have now released our implementation in an anonymized GitHub repository (https://anonymous.4open.science/r/RankFlow-EF7F/). The repository includes environment setup, training and evaluation scripts for experiments reported in the paper, together with configuration files and instructions to reproduce the benchmark results.

---

> ### Author Response · Authors · 2025-11-21
>
> **Question 1** **- Source of structures**
>
> In all our experiments, all protein structures are predicted by AlphaFold2. We have now clarified in Section 4.1 that among these methods, ProtSSN, SaProt, DePLM, and our RankFlow model are sequence+structure approaches that use both amino acid sequences and 3D structures, ESM-IF and ProteinMPNN use structures as inputs, whereas all other methods operate on sequence information only.
>
> **Question 2** **- Discussion about how to choose hyperparameters**
>
> Please refer to the response to the **Weakness** **1 - Discussion of hyperparameters**.
>
> **Question 3** **- Discussion about the gate**
>
> The gate is very data-efficient because it learns a position-wise gating vector of length $L$. In practice, we found that the gate becomes stable even when each positive/negative set contains on the order of 50-150 variants. The nature of the assay does influence how the gate behaves, but not in a way that requires special tuning. When the signal is concentrated on a subset of positions, the gate assigns high weights to those hotspots and suppresses positions that are uninformative across mutants. For assays with more diffuse or noisy signal, the gate acts more like a soft regularizer, it mildly downweights globally uninformative positions while leaving overall performance very close to the variant without gating.
>
> **Question 4** **- Learnable embeddings**
>
> By “applied only at the mutated positions” we mean that the additional learnable embeddings are injected only at residue indices where the mutant differs from the wild type, and not at every position in the sequence. This does not imply that the effect of mutations is restricted to those positions. Instead, these mutation-specific embeddings serve as localized signals that are subsequently propagated across all positions through the conditional flow network.
>
> **Question 5** **- Additional clarification regarding the modeling of the energy function in Eq. 7**
>
> The energy is designed with two complementary signals: (i) a global standardized fitness term $\tilde{y}\_i$, which captures how “good” a mutant is relative to the overall assay distribution, and (ii) a local deviation term $\frac{\tilde{y}_i - \bar{\tilde{y}}_i}{\sqrt{s_i}}$, which measures how much mutant $i$ deviates from the expected behavior of nearby mutants (i.e., mutants with similar substitution patterns). This identifies mutants that are unusually strong or weak relative to their mutational neighborhood. The mixing coefficient $\lambda$ in Eq. 7 balances these global and local signals. We have expanded the explanation of Eq. 7 (Lines 234–238) to clarify how the energy function is modeled.
>
> **Minor comments**
>
> We thank the reviewer for these helpful suggestions. We have carefully revised the manuscript to correct the misuse of \citet and \citep. The acronym “PLMs” is now defined only once at its first occurrence, and we have checked the text for consistency in the use of hyphens (e.g., “deep mutational scanning”) and number formatting (commas for thousands). Figure 4 has been updated with a larger font size and higher resolution to improve readability in printed form, and typos have been corrected.

---

> > ### Comment · Reviewer_3v7D · 2025-11-25
> > **Response to Authors**
> >
> > Thank you for addressing my comments. I have raised my score. I encourage the authors to open-source the method on Github.

---

> > > ### Author Response · Authors · 2025-11-26
> > >
> > > We sincerely thank the reviewer for the constructive feedback and for raising the score. We appreciate the suggestion regarding open-sourcing and plan to release our implementation on GitHub after the review period to facilitate reproducibility and future work.

---

### Official Review · Reviewer_PgXb · 2025-11-01

**Soundness:** 3
**Presentation:** 3
**Contribution:** 2
**Rating:** 4
**Confidence:** 4

**Summary:**

The goal of this work is to develop a **fitness predictor** to assess the effects of mutations on a given protein sequence (wild type). This is a central problem in computational biology, with applications in both **genomics** and **computational protein design**.

The authors propose a new computational procedure called **RankFlow**, which employs **flow matching** to learn a distribution in the latent space of a protein language model (PLM). This distribution is *tilted* toward mutants that are both high in fitness and anomalous relative to local substitution trends. Samples from this learned distribution are then passed through a PLM head and subsequently used for fitness prediction.

Results on relevant benchmarks show that RankFlow performs well compared to multiple existing methods.

**Strengths:**

- Good performance on multiple relevant benchmarks.
- The method can be pre-trained on multiple DMS assays, thereby benefiting from previously observed cross-protein transfer effects [1]. I believe this is the main strength of the proposed approach.

**Weaknesses:**

- The evaluation is missing two important baselines: [2] and [3].
- The evaluation does not include **out-of-distribution (OOD)** splits from **ProteinGym**.
- The authors do not propose any means for **uncertainty quantification (UQ)**, which limits the applicability of this predictor for **Bayesian Optimization**.
- Additional **ablations** on architectural choices would strengthen the work. For example:
  - What happens if the flow-matching module is replaced with a simple MLP layer?
  - What happens if the ranking loss is replaced with other commonly used losses?
  - How does this method compare with an approach similar to **CPT-1** [1]?

**Questions:**

1. Can diffusion be used to provide **uncertainty quantification**? It would be interesting to compare this form of UQ with other approaches [3, 4].
2. The authors claim that previous methods do not capture **epistasis**, but PLM embeddings can, in principle, model epistasis. Can the authors provide a comparative analysis of their model on multiple mutants versus other methods?
3. The notation is confusing in places. For example:
   - In **Figure 2**, the unified representation is denoted as *F*, but *F* never appears in the flow-matching equations—can the authors clarify this?
   - In **Equation (3)**, what are \(\dot{y}\) and \(\dot{\sigma}\)?
4. Can the authors clearly explain how the **datasets are split**? Do they use the random split for ProteinGym?
5. Can the authors clearly define how **fitness predictions** are generated according to their method in the main text?
6. Is the method sensitive to any **flow-matching hyperparameters**? How does randomness in the flow-matching process affect predictions?

---

[1] Jagota, M., Ye, C., Albors, C., Rastogi, R., Koehl, A., Ioannidis, N., & Song, Y. S. (2023). *Cross-protein transfer learning substantially improves disease variant prediction.* **Genome Biology**, 24(1), 182.

[2] Groth, P. M., Kerrn, M., Olsen, L., Salomon, J., & Boomsma, W. (2024). *Kermut: Composite kernel regression for protein variant effects.* **Advances in Neural Information Processing Systems**, 37, 29514–29565.

[3] Ronen, O., Zhao, A. Y., Boger, R., Ye, C., & Yu, B. (2025). *Stabilizing protein fitness predictors via the PCS framework.* In *The Exploration in AI Today Workshop at ICML 2025.*

[4] Greenman, K. P., Amini, A. P., & Yang, K. K. (2025). *Benchmarking uncertainty quantification for protein engineering.* **PLOS Computational Biology**, 21(1): e1012639.

---

> ### Author Response · Authors · 2025-11-21
>
> Thank you very much for your insightful comments and for recognizing cross-protein transfer as a key strength of our approach.
>
> **Weakness** **- Missing baselines [2] and [3] and lacking out-of-distribution (OOD) splits from ProteinGym**
>
> We have added Tables 2 & 3 in the revised manuscript to include comparisons with baselines Kermut [2] & PCS [3] under the Random, Contiguous and Modulo splits from ProteinGym.
> The new Table 2 shows that RankFlow is more stable across distribution shifts than other baselines, achieving the highest average performance over three evaluation modes compared to the second-best method Kermut (0.669 vs. 0.655). Specifically, RankFlow attains the highest Spearman scores on both the Random and Modulo splits (0.786 vs. 0.744 and 0.635 vs. 0.631, respectively). The Modulo split partitions positions into periodic subsets. RankFlow's conditional-flow formulation appear to leverage this dispersed contextual information more effectively than kernel-based regression. Kermut achieves the best performance on the Contiguous split with a Spearman correlation of 0.591, while our method attains a closely comparable score of 0.589.
>
> | Model           | Random | Modulo | Contiguous | Avg.  |
> |-----------------|--------|--------|------------|-------|
> | ProteinNPT      | 0.741  | 0.588  | 0.529      | 0.619 |
> | Kermut  [2]      | 0.744  | 0.631  | 0.591      | 0.655 |
> | RankFlow (Ours) | 0.786  | 0.635  | 0.589      | 0.669 |
>
> In Table 3, RankFlow outperforms all single supervised baselines, including Kermut, as well as the Stable CNN variant from PCS [3] in terms of average performance.  It is important to note that PCS trains dozens of predictors over 36 different protein representations and zero-shot scores, and uses the PCS pred-check procedure to select and ensemble stable predictors. As such, PCS is conceptually distinct and complementary to our work: its stronger performance under the Contiguous and Modulo schemes arises from ensembling and model selection rather than improvements to a single supervised architecture. In this sense, our RankFlow could naturally serve as one of the base predictors within the PCS framework.
>
> | Model           | Contiguous Spearman | Contiguous ρ_uncertainty | Modulo Spearman | Modulo ρ_uncertainty | Random Spearman | Random ρ_uncertainty |
> |-----------------|---------------------|---------------------------|-----------------|----------------------|-----------------|----------------------|
> | CNN             | 0.344±0.013         | 0.019±0.006              | 0.344±0.013     | 0.019±0.006          | 0.365±0.012     | 0.013±0.006          |
> | Stable CNN  [3]    | 0.492±0.010         | 0.129±0.008              | 0.492±0.010     | 0.129±0.008          | 0.509±0.010     | 0.133±0.008          |
> | Bayesian Ridge  | 0.422±0.014         | 0.008±0.004              | 0.422±0.014     | 0.008±0.004          | 0.693±0.013     | 0.005±0.004          |
> | Stable BayesR [3]  | 0.597±0.011         | 0.153±0.008              | 0.597±0.011     | 0.153±0.008          | 0.755±0.011     | 0.139±0.007          |
> | Kermut          | 0.606±0.012         | 0.110±0.008              | 0.606±0.012     | 0.110±0.008          | 0.758±0.012     | 0.106±0.008          |
> | Stable Kermut [3]  | 0.667±0.010         | 0.164±0.010              | 0.667±0.010     | 0.164±0.010          | 0.785±0.010     | 0.184±0.008          |
> | RankFlow (Ours) | 0.589±0.009         | 0.135±0.009              | 0.635±0.010     | 0.132±0.008          | 0.786±0.012     | 0.165±0.006          |
>
> **Weakness** **- Uncertainty quantification (UQ)**
>
> We thank the reviewer for pointing out the UQ for Bayesian Optimization. We’d like to clarify that our proposed RankFlow is fully compatible with established UQ techniques. In the revision, we have adopted the hybrid uncertainty estimation used in ProteinNPT, combining MC-Dropout and batch-resampling to obtain epistemic uncertainty scores. This approach is model-agnostic and does not require any changes to our model architecture. We have included UQ results following this approach in Table 3. As can be observed, our method not only achieves strong predictive accuracy but also delivers robust uncertainty quantification on par with PCS-stabilized baselines. We also notice that the ρ_uncertainty values reported for all methods in PCS [3] are identical under the Contiguous and Modulo splits including standard errors, which is unlikely given the different ProteinGym splits. We therefore rely primarily on the Random-split PCS results. Under this setting, our method attains a ρ_uncertainty of 0.165 ± 0.006, which is lower than Stable Kermut (0.184 ± 0.008) but higher than the other PCS methods (e.g., 0.139 ± 0.007 for Stable Bayesian Ridge). This indicates that our approach provides informative and well-correlated uncertainty estimates, even without the additional stabilization machinery introduced by the PCS framework.

---

> > ### Author Response · Authors · 2025-11-21
> >
> > **Weakness** **- Additional ablations**
> >
> > (1) Replacing the conditional flow module with an MLP.
> > This scenario is already represented in our comparison in Table 1. Specifically, the results reported for ESM2 come from DePLM, where a trainable MLP is attached to the pretrained ESM2. Across ProteinGym assays, replacing the flow with an MLP causes a substantial average drop of 16% in Spearman correlation (0.654 vs. 0.784). This supports our claim that the conditional flow architecture itself, rather than just additional capacity, is important for reshaping PLM embeddings into a property-aligned representation.
> >
> > (2) Ablation of the ranking loss.
> > Our original submission already included a loss ablation where we replaced the ranking loss with kinetic regularization (Line 979 in Table 6, now Table 8), which consistently reduced performance across assays. In response to this comment, we have additionally evaluated a standard pairwise hinge ranking loss (Line 980, Table 8), which underperforms our ranking loss by 2.82 Spearman points. Conceptually, our rank-consistent loss provides denser and more stable gradients, captures global ranking structure rather than disjoint pairs, and handles continuous fitness values more appropriately.
> >
> > (3) Ablation of linear regressor from CPT-1.
> > We performed additional ablation on eight representative ProteinGym assays by replacing our flow module with a linear regressor similar to CPT-1. Across these assays, the linear regressor achieves a Spearman correlation of 0.425, whereas our flow module achieves 0.691. This gap indicates that our conditional flow architecture captures richer, non-linear fitness structure than a CPT-1–style linear baseline. The eight assays are: BLAT\_ECOLX\_Jacquier\_2013, CALM1\_HUMAN\_Weile\_2017, DLG4\_RAT\_McLaughlin\_2012, DYR\_ECOLI\_Thompson\_2019, P53\_HUMAN\_Giacomelli\_2018, REV\_HV1H2\_Fernandes\_2016, RL40A\_YEAST\_Roscoe\_2013, and TAT\_HV1BR\_Fernandes\_2016.
> >
> > **Question 2** **- Clarification on epistasis modeling and comparison analysis on multiple mutants**
> >
> > Our original statement “Second, existing methods overlook multi-mutation interactions (epistasis) …” was intended to refer to fitness prediction models, not to PLM embeddings themselves. We agree that PLM embeddings do capture epistasis, our point is that some fitness prediction methods (e.g., One-Hot Encodings, DePLM) assume independent site-wise contributions, and do not explicitly parameterize or model higher-order interactions. To make this precise, we have revised the sentence to: “Second, existing fitness prediction models (e.g., One-Hot Encodings, DePLM) based on position-wise features, overlook multi-mutation interactions (epistasis) and….”.
> >
> > We now compare our method with ProteinNPT on a set of multi-mutant DMS assays (Figure 5, Appendix B.3). Across these assays, RankFlow attains higher Spearman correlation on the large majority of landscapes. The largest improvements (up to 0.1 absolute Spearman) occur on the more challenging, highly combinatorial landscapes (e.g. SPG1_STRSG_Wu_2016 and GCN4_YEAST_Staller_2018), indicating that the proposed flow-based, property-aware flow can extract and align epistatic signal from PLM representations more effectively than ProteinNPT.
> >
> > **Question 3** **- Notation**
> >
> > In our implementation, the unified representation $\boldsymbol{F}$ is not used directly in the flow-matching equations, instead, it enters the flow-matching equations only as conditioning. Concretely, $\boldsymbol{F}$  is combined with the steering gate $\boldsymbol{g}$  to form the condition $\boldsymbol{C} =[\boldsymbol{F},\boldsymbol{g}]$, and the flow field is defined as $\boldsymbol{v} _t(\boldsymbol{h} \mid \boldsymbol{F}; \theta)$. Thus, $\boldsymbol{F}$  influences the flow entirely through $\boldsymbol{C}$, which controls how the sequence embedding is transformed over time. In the revised manuscript we have clarified this by explicitly stating that we first compute the fused representation $\boldsymbol{F}$, then form $\boldsymbol{C}$ =[$\boldsymbol{F}$,$\boldsymbol{g}$], and define the flow field as $\boldsymbol{v} _t(\boldsymbol{h} \mid \boldsymbol{F}; \theta)$ (Lines 326-327).
> >
> > $\dot{\mu}_t$ and $\dot{\sigma}_t$ are both the derivative of $\mu_t$ and $\sigma_t$ with respect to time t.   We have updated this in the revised manuscript (Line 207).
> >
> > **Question 4** **- Datasets splits**
> >
> > Results from Tables 1 uses the Random split for ProteinGym, which was described in Appendix B.1 Details of Benchmark. In the revision, we have clarified the Random scheme in Section 4.1 when describing Table 1. And we have added Table 2 to report results of different methods under all three ProteinGym evaluation schemes: Random, Modulo, and Contiguous.

---

> > > ### Author Response · Authors · 2025-11-21
> > >
> > > **Question 5** **- Fitness predictions**
> > >
> > > Following ProteinGym, the fitness predictions are calculated by taking the log-ratio of the sequence probabilities between the mutant and wild-type sequences $\frac{p(\mathbf{x}^{\mathrm{mt}})}{p(\mathbf{x}^{\mathrm{wt}})}$.
> > >
> > > **Question 6** **- Sensitivity to flow-matching hyperparameters and randomness in flow-matching process**
> > >
> > > In our experiments, the method is not particularly sensitive to flow-matching hyperparameters. We follow the standard flow-matching setup, where the main choices are: (i) the distribution over time $p(t)$, which we use a fixed uniform distribution [0,1], and (ii) the time scheduling, where we adopt a cosine schedule instead of a linear one after ablation. We found that switching between linear and cosine schedules has only a minor effect on absolute performance and does not change the relative ranking to DePLM and ProteinNPT.
> > >
> > > Regarding randomness, the stochasticity in flow matching appears only during training, through the random sampling of time points $t$ and noise. At test time, unlike the unconditional generative setting in flow matching, we treat the PLM embedding of a sequence as the source state and obtain predictions by numerically solving the ordinary differential equation (ODE) from $t$=0 to 1 with a fixed step schedule, so the mapping from input sequence to predicted fitness is deterministic.

---

> ### Comment · Reviewer_PgXb · 2025-11-22
> **Thanks for the detailed response.**
>
> I thank the authors for their detailed response.
>
> With these new comparisons, I am now convinced that RankFlow is a strong method, which utilizes cross-protein transfer learning for fitness prediction. I will therefore raise my score.
>
> I do have a few other questions for the authors:
> * When you provide UQ as proposed by ProteinNPT, can you explain why would there be a batch effect? Their method had cross data point attention. Does your method have some dependency on the particular batch, or does the randomness come from the diffusion process. If the former, it would be insightful to disentangle the two, if it is the latter, please clarify in writing.
>
> * I would also be curious to understand which UQ method is stronger, MC-dropout or the batch/diffusion.

---

> > ### Author Response · Authors · 2025-11-24
> >
> > We appreciate the reviewer for the encouraging feedback on our previous response and for these follow-up questions. For RankFlow, there is no dependency on the particular test batch, the model does not use cross-data point attention, and predictions for a given variant depend only on that variant's input. The randomness in our UQ for RankFlow comes solely from the stochastic forward passes used for MC-Dropout and from training-data resampling, where we rebuild the inference batch five times with different random subsets of the training set following ProteinNPT. It does not come from the diffusion process or from other variants in the test batch. In our experiments, MC-Dropout is the primary driver of the uncertainty signal, while batch resampling contributes noticeably only on low-data or particularly difficult assays.

---

### Official Review · Reviewer_sQ9J · 2025-11-01

**Soundness:** 2
**Presentation:** 2
**Contribution:** 3
**Rating:** 4
**Confidence:** 3

**Summary:**

This paper presents RankFlow, a property-aware conditional flow model for protein optimization. It refines PLM embeddings into property-aligned representations. It captures complex multi-mutation interactions through an energy-guided flow and a Property-Guided Steering Gate (PSG). The Rank-Consistent Conditional Flow Loss (RC² Loss) enforces correct mutant ranking and ensures out-of-distribution generalization. The model achieves superior performance on various benchmarks.

**Strengths:**

Unlike existing methods that simply add the mutational effect of an individual site, RankFlow uses energy-guided conditional transport to capture complex multi-mutation interactions. In addition, optimizing the relative ordering of mutants provides better generalization to unknown protein families.

**Weaknesses:**

The model achieves better performance than previous models. However, there could be more detailed experimental setup descriptions, especially as the setup is not identical to the standard ProteinGym benchmark. The author could also discuss more on other supervised models, such as ProteinNPT and Metalic.

**Questions:**

1. The ProteinGym benchmark has a standard evaluation pipeline for unsupervised and supervised fitness prediction tasks. What is the difference between the evaluation procedure this paper follows and the standard ProteinGym evaluation pipeline?

2. Are the unsupervised methods and supervised methods evaluated on the same set of protein sequences in Tables 1 and 2?

---

> ### Author Response · Authors · 2025-11-21
>
> Thank you for reviewing our paper and for highlighting the superior generalization ability of our proposed method as a key strength.
>
> **Weakness** **- Experimental setup**
>
> As stated in our manuscript, our training and evaluation procedures follow the ProteinGym supervised benchmark (217 assays) under the Random five-fold scheme (Lines 750-768, Appendix B.1). We have now also made this Random scheme explicit in Section 4.1 when describing Table 1. The only deviation from the full ProteinGym setup is that, following prior PLM-based work such as DePLM, we exclude assays whose wild-type sequence length exceeds 1,024 residues. This results in 201 DMS datasets and ensures a fair comparison with DePLM. All other aspects, including fold construction and evaluation procedure, strictly follow the ProteinGym benchmark.
>
> We have also added Table 2 in the revised manuscript to include comparisons with baselines under the Random, Contiguous and Modulo splits from the full ProteinGym setup. The new Table 2 shows that RankFlow is more stable across distribution shifts than other baselines, achieving the highest average performance over all three evaluation modes compared to the second-best method Kermut (0.669 vs. 0.655). Specifically, RankFlow attains the highest Spearman scores on both the Random and Modulo splits (0.786 vs. 0.744 and 0.635 vs. 0.631, respectively). The Modulo split partitions positions into periodic subsets. RankFlow's conditional-flow formulation appear to leverage this dispersed contextual information more effectively than kernel-based regression. Kermut achieves the best performance on the Contiguous split with a Spearman correlation of 0.591, while our method attains a closely comparable score of 0.589.
>
> | Model           | Random | Modulo | Contiguous | Avg.  |
> |-----------------|--------|--------|------------|-------|
> | ProteinNPT      | 0.741  | 0.588  | 0.529      | 0.619 |
> | Kermut          | 0.744  | 0.631  | 0.591      | 0.655 |
> | RankFlow (Ours) | 0.786  | 0.635  | 0.589      | 0.669 |
>
> **Weakness** **- Discussion of ProteinNPT and Metalic in Related Work**
>
> Compared with ProteinNPT and Metalic, RankFlow offers a more computationally efficient approach while maintaining strong performance across dense and label-sparse DMS settings. ProteinNPT is a conditional pseudo-generative non-parametric Transformer that jointly embeds protein sequences and labels and is particularly strong in label-scarce, multi-task DMS settings. While ProteinNPT is a strong supervised baseline, its tri-axial attention and non-parametric conditioning over labeled sequences make it computationally and memory intensive. For example, training ProteinNPT on a single dataset with a protein of length 267 and 5,001 mutants already requires around 5.85$\times10^9$ MACs (Table 9, Lines 1028-1033), and the cost grows quickly with sequence length and the number of mutants. Metalic is a meta-learning approach that trains a sequence-only in-context regressor over multiple fitness prediction datasets. However, it assumes access to a large distribution of related datasets, requires 2-8 days of training on an A100-80GB GPU, and excludes proteins longer than 750 amino acids due to memory limits. In contrast, RankFlow is trained directly on individual landscapes, learns a property-aware conditional flow guided by an explicit energy function and rank-consistent flow-matching objective, and uses a lighter architecture without meta-training or large in-context batches. In our experiments, RankFlow is trained for 10–50 epochs per dataset and typically completes in under one hour on a single A100 GPU. This makes RankFlow practical for dense multi-mutant campaigns and scenarios, while still capturing the geometry of the fitness landscape that underpins accurate ranking and design. We have expanded Section 2 to include these discussions (Lines 129-136).
>
> **Question 1** **- Evaluation procedure**
>
> Please refer to the response to the **Experimental setup** section.
>
> **Question 2** **- Evaluation of unsupervised methods and supervised methods**
>
> All unsupervised and supervised methods are evaluated on the same set of protein sequences in Tables 1 and 2. Specifically, For Table 1, all models are trained and evaluated on the ProteinGym supervised benchmark using the Random five-fold scheme.
> For the Table 2 (now Table 4 in the revised manuscript), each DMS dataset serves as a test dataset in turn. As stated in Lines 470-472, for a given test dataset, we randomly select up to 40 datasets from the same functional category as the training dataset while enforcing sequence similarity between training and test sequences below 50% to prevent data leakage.

---

> ### Comment · Reviewer_sQ9J · 2025-11-24
>
> Thank you for the author's response. I have raised my score.

---

> > ### Author Response · Authors · 2025-11-24
> >
> > We appreciate your review and the time you took to read our rebuttal and adjust your score. Your comments were very helpful in improving the clarity and presentation of our work.

---

### Author Response · Authors · 2025-11-29

Dear PCs, Senior ACs, and ACs,

We would like to thank the AC and reviewers and briefly summarise the review and discussion process for our submission. We have successfully addressed all concerns raised by Reviewers sQ9J, PgXb, and 3v7D, who explicitly wrote that our rebuttal resolved their concerns and raised their scores from 4 to 6, 4 to 6, and 6 to 8, respectively.

We also addressed all the comments from Reviewer XW2e, in particular by providing the requested statistical test result, which confirms that our proposed RankFlow achieves statistically significant improvements. Although the discussion closed before we could receive further feedback from Reviewer XW2e, we are confident that the quantitative evidence provided fully resolves the final concern.

Regarding the reviews and how they evolved during our rebuttal and the discussion:

- **Reviewer sQ9J**

In the initial review, he/she raised two questions about the experimental setup. After reading our rebuttal, he/she wrote that his/her concerns had been resolved and raised the score from 4 to 6 on 24th Nov.

- **Reviewer PgXb**

In the initial review, he/she highlighted that "The method can be pre-trained on multiple DMS assays, thereby benefiting from previously observed cross-protein transfer effects. I believe this is the main strength of the proposed approach." After the rebuttal and discussion, he/she stated that he/she is now convinced that RankFlow is a strong method that utilises cross-protein transfer learning for fitness prediction and raised the score from 4 to 6 on 23th Nov.

- **Reviewer 3v7D**

In the initial review, he/she wrote that "Overall, the paper contributes a new flow-based framework for protein fitness prediction that could be a contribution to epistasis modeling." and had several questions about hyperparameters and implementation details. After our rebuttal, he/she stated that his/her comments were addressed and raised the score from 6 to 8 on 26th Nov.

- **Reviewer XW2e**

In the initial review, he/she wrote that our idea "appears novel, original and well-motivated" and raised several major concerns: the motivation for a generative rather than regression formulation, the discrepancy with results from the ProteinGym website, and details about training data and parameter count comparison. In our rebuttal, we provided a clearer high-level intuition for RankFlow and then explicitly pointed to the relevant sections of the submitted manuscript where the evaluation protocol, reported numbers, baseline choices, training data, and parameter counts are already specified, adding brief clarifications to avoid misunderstanding. During the discussion, he/she stated that he/she appreciated the rebuttal and posed a follow-up question about the statistical test between RankFlow and DePLM, for which we provided the detailed per-assay statistical comparison, showing that RankFlow’s improvements over DePLM are statistically significant.

Thank you very much for your time and for overseeing the review process.

Best regards,
Authors

---

### Meta-Review · Area_Chair_UznQ · 2026-01-04

**Summary:**

This paper introduces a new generalizable protein fitness prediction framework, whereby PLM-embeddings are adapted using a fitness-tilted flow matching model, where a novel ranking-score and energy function is used to condition (tilt) the flow matching model. The authors show their method is SOTA on protein-gym benchmarks against strong baselines.

Strengths:
- Introduces a fitness prediction framework that explicitly accounts for epistatic mutation effects (as opposed to, e.g. using a regression head on average-pooled PLM embeddings), and is trained on multiple DMS datasets for this purpose
- Multiple innovations for improving fitness prediction presented (e.g. rank prediction as opposed to value prediction, fitness-tilted FM process, property-aware steering gate)
- Good empirical performance on a wide range of benchmarks


Weaknesses:
- Standard OOD hold-out sets from protein-gym experimental setup not used, no clear separation between zero-shot and trained methods
- Some important baselines missing
- Important ablations missing that would help to justify complexity/architectural choices and hyperparameter settings
- Discussion/motivation missing for why generative flow-matching required for a discriminative task.

This paper is solving a topical and relevant problem and presents a number of interesting new methodological contributions. As far as I can tell, the authors address all of the major concerns from the reviewers, which justify score increases. As such, I will recommend this paper for acceptance into ICLR.

**Reviewer Concerns:**

Concerns addressed:
- Standard protein-gym setup is used, and OOD hold-out sets added, with RankFlow achieving good performance.
- Zero-shot and supervised methods clearly delineated
- Baselines that were missing have been added, and RankFlow still achieve SOTA performance.
- Validated adding predictive uncertainty to RankFlow by using MC-dropout+batch resampling
- Ablations added to the appendix that justify architectural choices, e.g. flow-matching vs MLP, structure embeddings removed, etc
- Information added on hyperparameter selection procedures
- Discussion and figure (1) added to clarify motivation and function of flow-matching vs. regression head PLM adaptation
- Statistical tests of RankFlow vs DePLM performance


Concerns unaddressed:
- I believe all major concerns have been addressed

**Reviewer Scores:**

sQ9J: 4 -> 6 as stated by reviewer, and I think this score increase is justified

PgXb: 4 -> 6 as stated by reviewer, and I think this score increase is justified

3v7D: 6 -> 8 as stated by reviewer, and I think this score increase is justified

XW2e: 2 -> 4 or 6 I believe the major concern were all addressed

New score: 6 to 6.5

---

### Decision · Program_Chairs · 2026-01-26

Accept (Poster)